



# Numerical modeling of the dynamics of Mer de Glace glacier, French Alps: comparison with past observations and forecasting of near future evolution.

Vincent Peyaud[1], Coline Bouchayer[1,2], Olivier Gagliardini[1], Christian Vincent[1], Fabien Gillet-Chaulet[1], Delphine Six[1], and Olivier Laarman[1]

[1]Univ. Grenoble Alpes, CNRS, IRD, Grenoble INP, IGE, 38000 Grenoble, France
[2]Department of Geosciences, University of Oslo, 0316 Oslo, Norway

*Correspondence to:* Vincent Peyaud (vincent.peyaud@univ-grenoble-alpes.fr)

**Abstract.**

All alpine glaciers are shrinking and retreating at an accelerating rate in a warming climate. Glacier modeling is required to assess the future consequences of this retreat on water resources, the hydropower industry and risk management. However, the performance of such ice flow modeling is generally difficult to evaluate because of the lack of long-term glaciological observations. Here, we assess the performance of the Elmer/Ice full-Stokes ice flow model using the long dataset of mass balance, thickness change, ice flow velocity and snout fluctuation measurements obtained between 1979 and 2015 on the Mer de Glace (Mont Blanc area). Ice flow modeling results are compared in detail to comprehensive glaciological observations over four decades including both a period of glacier expansion and a long period of decay. To our knowledge a comparison to data at this detail is unprecedented. We found that the model accurately reconstructs the velocity and elevation variations of this glacier despite some discrepancies that remain unexplained. The calibrated and validated model was then applied to simulate the future evolution of Mer de Glace from 2015 to 2050 using 26 different climate scenarios. Depending on the climate scenarios, this glacier, the largest in France, could retreat by 2 to 5 km over the next three decades.

## 1 Introduction

Mountain glacier mass balances show a strong sensitivity to climate change and can thus be used to assess the impact of climate change in remote areas (Oerlemans, 2001; Zemp et al., 2019). During the 20th century, all alpine glaciers showed a strong recession (Zemp and Frey, 2015). This observed trend is expected to continue in the future under a warming climate (IPCC, 2019) with important impacts on watershed hydrology (Huss and Hock, 2018; Brunner et al., 2019), tourism and hydropower resources (e.g. Welling et al., 2015; Stewart et al., 2016), accompanied by the emergence of new risks (e.g. Kääb et al., 2018) and sea-level rise (Bamber et al., 2018; Marzeion et al., 2018; Parkes and Marzeion, 2018). Properly assessing these future impacts requires the development of modeling tools capable of describing the processes driving these glacier changes.

Numerical ice flow models with different degrees of complexity have been developed to forecast glacier fluctuations. The first studies (since Haeberli and Hölzle, 1995) focused on an empirical approach in which ice dynamics were not taken into





account explicitly and glacier evolution was based on parameterization calibrated either on equilibrium-line altitude (ELA) model (e.g. Zemp et al., 2006), extrapolation of observed geometry changes (e.g. Huss et al., 2008; Huss, 2012; Huss and Hock, 2018) or volume and length–area scaling (e.g. Marzeion et al., 2012; Radić et al., 2014). Process-based model were also developed to take into account simple dynamics (e.g. Clarke et al., 2015; Zekollari et al., 2019; Maussion et al., 2019).

These studies suggest a glacier volume loss from 65% to 94% in the Central Europe by the end of the century depending on the climate scenario (Zekollari et al., 2019; Hock et al., 2019; IPCC, 2019). However, the Fourth IPCC Assessment Report (Solomon et al., 2007) and other studies (e.g. Vincent et al., 2014) emphasize the need for a new generation of glacier models that accurately describe the ice flow dynamics to correctly forecast individual glacier evolution. Today, such three-dimensional physical models are widely available. Indeed, with the increase in the performance of computational resources, running a model

describing the complex three-dimensional geometry of a whole glacier has become much more affordable (e.g. Le Meur and Vincent, 2003; Zwinger et al., 2007; Gagliardini et al., 2011; Gilbert et al., 2014; Jouvet and Funk, 2014; Réveillet et al., 2015; Farinotti et al., 2017; Gilbert et al., 2018). Among such models, Elmer/Ice (Gagliardini et al., 2013) has already been used for a number of glacier applications (e.g. Zwinger et al., 2007; Gagliardini et al., 2011; Gilbert et al., 2014; Réveillet et al., 2015; Gilbert et al., 2018) and will be used for this study.

However, very few glacier datasets are available to make a detailed comparison between observed and modeled fluctuations at the multi-decadal scale. The Mer de Glace glacier offers a rare opportunity to compare state-of-the-art model results with a large dataset containing observed thickness changes, ice flow velocities and snout fluctuations over a nearly continuous 40-year period thanks to the GLACIOCLIM observatory monitoring program (Vincent, 2002; Vincent et al., 2014; Berthier et al., 2004, 2005, 2014; Berthier and Vincent, 2012). In addition, running simulations on this glacier provides the opportunity to

simulate complex ice dynamics as the glacier presents a large expansion before the eighties followed by a rapid retreat over three decades. This dynamics make it necessary to take into consideration the delay in the glacier response to climatic forcing.

   In this paper, the performance of the Elmer/ice ice-flow model is first assessed in terms of its ability to reconstruct these past multi-decadal fluctuations. A thorough comparison makes it possible to explore the sources of discrepancies between the reconstruction and the observations. In a second step, prognostic simulations are performed to simulate the evolution of Mer

de Glace glacier until 2050 under different climate scenarios.

## 2   Study site and glaciological data

Mer de Glace ($45°55'$ N, $6°57'$ E), the largest glacier in the French Alps, covers an area of $32 \ \mathrm{km}^2$. It is located in the Mont Blanc massif (Fig. 1) and is monitored as part of the GLACIOCLIM observatory (https://glacioclim.osug.fr/). The maximum elevation of its upper accumulation area reaches $4300 \ \mathrm{m}$ a.s.l. From this accumulation region, the ice flows rapidly through a

narrow, steep portion (an icefall between 2700 and $2400 \ \mathrm{m}$ a.s.l.) before feeding the lower, $7 \ \mathrm{km}$ long part of the glacier down to a front located at $1534 \ \mathrm{m}$ a.s.l. in 2018. As shown in Fig. 1, Leschaux glacier is the only active tributary of Mer de Glace glacier.





Several surface Digital Elevation Models (DEM) are available for different time in the past. The first map was produced by Vallot (1905) using the classical topographic method in 1905. Another DEM was made by Institut Géographique National (IGN) in 1979 and two by the Laboratory of Glaciology of Grenoble in 2003 and 2008 using aerial photographs (Vincent et al., 2014). A surface velocity field was derived from SPOT 5. Moreover, continuous field measurements have been performed

in the lower part of the glacier (below 2300 m a.s.l.) from a network of stakes maintained continuously since 1979 at four different elevations: the Tacul, Trélaporte, Echelets and Montenvers cross sections (see Fig. 1). Surface elevation has been measured systematically each year since 1979 along these four cross sections. Surface mass balance and annual surface velocity observations are also available at these cross sections although they are not continuous between 1979 and 1994, except for observations at the Tacul glacier cross section which are continuous over the whole period.

The bedrock topography was determined below 2300 m a.s.l. using mechanical borehole drillings, seismic soundings (Süustrunk, 1951; Vallon, 1961, 1967; Gluck, 1967) and radar measurements (2018, not published). Given the paucity of bedrock topography measurements in the upper part of the glacier (above the ice fall of Géant glacier) and the absence of measurements of bedrock topography for Leschaux glacier, the model domain is restricted to the lower part of the glacier from Tacul glacier down to the snout. We assume that the contribution of the Géant and Leschaux glaciers to the Mer de Glace glacier can be

represented as specified flux conditions on the boundary of the Mer de Glace model.

## 3   Ice flow model

Mer de Glace ice flow dynamics are modeled with the Elmer/Ice open-source finite-element model (Gagliardini et al., 2013). This model has been applied to simulate real and artificial mountain glaciers (e.g. Zwinger et al., 2007; Gagliardini et al., 2011; Gilbert et al., 2014; Réveillet et al., 2015; Farinotti et al., 2017; Gilbert et al., 2018). The main equations solved within

Elmer/Ice are summarized below. For more details regarding their numerical implementation, the reader can refer to Gagliardini et al. (2013) and the papers listed herein.

### 3.1   Field equations

We use Glen's law (Glen, 1955), a viscous isotropic nonlinear flow law, to link the deviatoric-stress tensor $\boldsymbol{\tau}$ to the strain-rate tensor $\boldsymbol{D}$:

$$\boldsymbol{\tau} = 2\mu\boldsymbol{D}\,, \tag{1}$$

where the ice effective viscosity $\mu$ is given by

$$\mu = \frac{1}{2}A^{-1/n}D_e^{1-1/n}\,. \tag{2}$$

In Eq. (2), $n$ is the Glen's exponent ($n = 3$), $D_e = 1/2tr(\boldsymbol{D}^2)$ the second invariant of the strain-rate tensor and $A$ is a rheological parameter with a constant value assuming temperate ice ($A = 5.0159e^{-24}$ Pa$^{-3}$s$^{-1}$). Indeed, the ice of the lower

part of the Mer de Glace glacier is most likely temperate (Lliboutry et al., 1962).





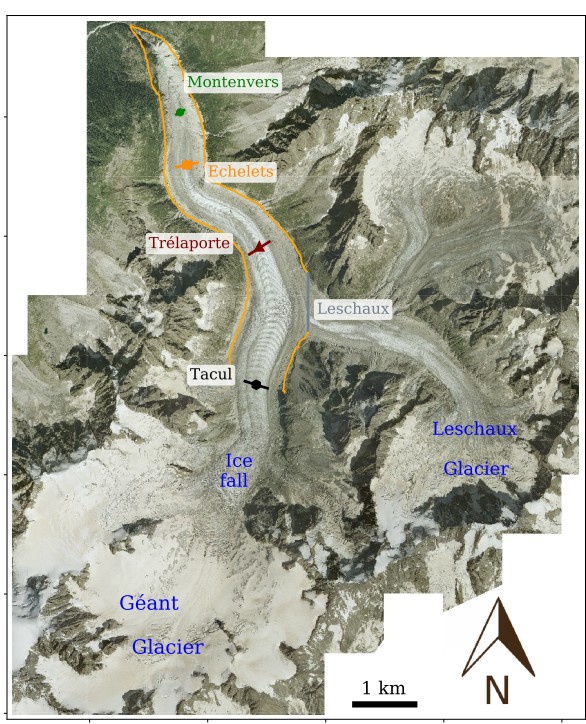

**Figure 1.** Map of Mer de Glace (orthophotoplan acquired in 2008 ©RGD74). Orange contour delimits the area modeled in this study. The location of the four cross sections (Tacul, Trélaporte, Echelets and Montenvers) and the Leschaux gate are indicated by the colored lines. The Tacul and Leschaux gates represent boundary gates where data is used to force the model whereas the three other profiles represent internal gates where data is used to validate the model.

The 3d velocity field $\boldsymbol{u} = (u, v, w)$ and $p$, the isotopic pressure, are solution of the Stokes equations that expresses conservation of momentum and conservation of mass for an incompressible fluid:

$$\nabla \boldsymbol{\tau} - \boldsymbol{\nabla} p - \rho \boldsymbol{g} = 0, \tag{3}$$

and

5  $$\nabla \boldsymbol{u} = 0, \tag{4}$$

where $\boldsymbol{g}$ is the gravity vector and $\rho$ the ice density. The upper surface of the glacier is a free surface of elevation $z_s$ (m) that evolves with time according to the kinematic equation:

$$\frac{\partial z_s}{\partial t} + u_s \frac{\partial z_s}{\partial x} + v_s \frac{\partial z_s}{\partial y} - w_s = b(z_s, t), \tag{5}$$

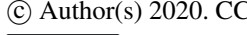



where the surface mass balance $b(z,t)$, in ice equivalent thickness ($\mathrm{m\,a^{-1}}$), is a function of surface elevation and time and $\boldsymbol{u_s} = (u_s, v_s, w_s)$ denotes the surface velocity vector. As the finite element mesh cannot have a null thickness, a lower limit of 1 m above the bedrock elevation is applied to $z_s$ in Eq. 5.

### 3.2 Boundary conditions

At the base, ice cannot penetrate into the bed so the velocity component normal to the bed is null. As Mer de Glace glacier is a temperate glacier, a certain amount of sliding on its bed is expected. A linear friction law relating the basal shear stress $\boldsymbol{\tau_b}$ to the basal velocity $\boldsymbol{u_b}$ is applied on the lower boundary:

$$\boldsymbol{\tau_b} + \beta \boldsymbol{u_b} = 0 \,. \tag{6}$$

The basal friction parameter distribution $\beta(x,y)$ is inferred using an inverse method described in Gillet-Chaulet et al. (2012).
This method relies on the computation of the adjoint of the Stokes system and the minimization of a cost function that measures the mismatch between modeled and observed velocities, using the surface topography and surface velocities measured in August 2003 (Berthier et al., 2004). The value of the basal friction parameter is kept constant in both past and future simulations.

On the upper surface, the surface mass balance, required in the glacier-free surface equation (Eq. 5), is derived either from observations when available or based on a positive degree-day (PDD) model forced by climate simulations for future. The two
methods are explained in detail in Section 4.1.

As the model domain does not cover the whole glacial catchment (see Fig. 1), ice flowing from the main accumulation area through the Tacul gate and from the tributary glacier (Leschaux) add two flux boundary conditions on the side of the domain. The flux coming from the upper part of the glacier through the Tacul gate boundary condition is imposed from observations (thickness and central horizontal velocity at the Tacul gate) from the past and the estimated flux in the future. A similar method
based on a flux is applied at the junction with Leschaux glacier. The implementation of an ice flux at these two gates for hindcast and forecast simulations differs slightly and is described in detail in Section 4.2.

### 3.3 3D mesh

The 3D mesh is constructed from the vertical extrusion of a 2D footprint that covers the 1905 extension of the glacier up to the Leschaux and Tacul boundaries. In this way, the glacier is allowed to advance up to its 1905 maximal extension. The 2D
footprint is meshed with unstructured triangular elements at a spatial resolution of $50$ m. Tests were performed with 6 and 10 extruded layers and do not show significant sensitivity to the vertical resolution. With six vertical layers, the resulting 3d mesh contains 32,784 nodes. The initial upper surface elevation at the mesh nodes is obtained from a cubic-spline interpolation (Haber et al., 2001) of the 1979 surface DEM. Bedrock elevation at the mesh nodes is obtained using natural-neighbour interpolation (Fan et al., 2005) of all available observations.





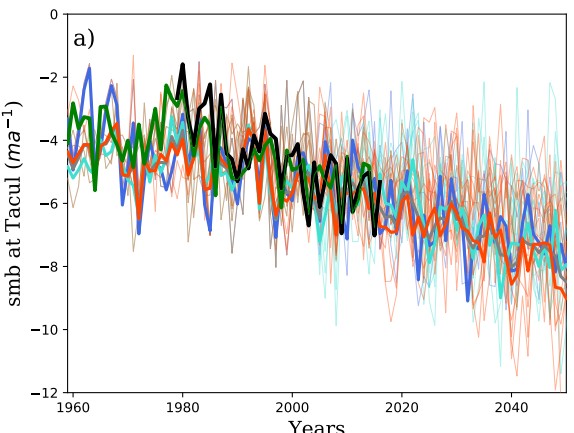
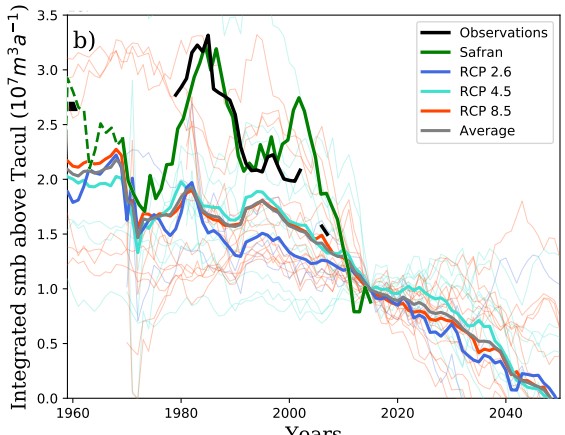

**Figure 2.** Evolution with time from 1960 to 2050 of a) the surface mass balances at Tacul gate and b) the integrated surface mass balance above Tacul gate. Observations are presented in black and values inferred from Safran in green. The others climate scenarios are plotted in dark blue (RCP 2.6), blue (RCP 4.5) and red (RCP 8.5); the average values for each scenario are highlighted by thick curves. Note that for the past period 1960-2015, the integrated surface mass balance above Tacul gate in b) does correspond to the flux at this gate and that "observations" are not from direct observations but are actually estimated from surface velocity and elevation following method used in Berthier and Vincent (2012). All integrated surface mass balances for the forecast simulations are normalized to the 2015 observed mass balance.

## 4    Methods

Our simulations cover the period 1979-2050. The hindcast simulation covers the period 1979-2015 for which annual surface mass balances, surface velocities and elevation changes are available yearly at the four cross sections of Tacul, Trélaporte, Echelets and Montenvers (Fig. 1). The dataset at the Tacul cross section is used to specify the flux on this artificial boundary

5   of the glacier domain, while the 3 others are used to evaluate the model over the hindcast period. For the forecast simulations from 2015 till 2050, results from climate simulations are used to evaluate the flux on the different boundary conditions of the glacier domain. This section describes in detail the respective boundary conditions for the two steps (hindcast and forecast) defined at the surface (surface mass balance) and at the Tacul and Leschaux gates (ice flux from the accumulation areas).

### 4.1    Surface mass balance

10  For the hindcast simulation, surface mass balance is derived from observations acquired during the historical period 1979-2015 (Six and Vincent, 2014, Fig. 2a). The surface mass balance is reconstructed according to an empirical relation linking the surface mass balance at a given elevation for a given year with the observed surface mass balance at the Tacul cross section gate for the same year, according to:

$$b(z_s, t) = k_b \left[ z_s(t) - z_{s\text{TAC}}(t) \right] b_{\text{TAC}}(t), \tag{7}$$



where $b_{\mathrm{TAC}}(t)$ is the annual surface mass balance measured (hindcast) or evaluated (forecast) at the Tacul altitude. The vertical mass balance gradient $k_b = \partial b / \partial z$ was estimated using the yearly surface mass balance measurements at the four profiles from 1995 to 2015 (see Fig. S1 in the Supplementary Material). A mean value of $k_b = 0.009\ \mathrm{m}^{-1}$ is obtained with a standard deviation of 0.002. Despite this strong variability from year to year (Fig. S1 in the Supplementary Material and Rabatel et al.,
2005), a constant mass balance gradient of $k_b = 0.009\ \mathrm{m}^{-1}$ is adopted for hindcast and forecast simulations.

For the future, the surface mass balance at Tacul gate in Eq. (7) is inferred from the temperature and the precipitation of 26 climate projections of the EURO-CORDEX program (Jacob et al., 2014) adapted to the Mont Blanc massif using the ADAMONT method (Verfaillie et al., 2017). The 26 scenarios used in this study cover the 3 IPCC scenarios RCP 2.6, 4.5 and 8.5 (see Table 1 in Supplementary Material).

A degree-day model (Braithwaite, 1995; Hock, 2003), known for its simplicity and relatively good performance, is used to evaluate the surface melt at Tacul gate from the modeled air temperature. Surface melting is proportional to the sum of positive degree-days (PDD, i.e. the sum of daily mean temperatures above the melting point over a given period of time) assuming different melt factors for snow and ice. These melt factors, here expressed in ice thickness equivalent, are $0.0048\ \mathrm{mK}^{-1}\mathrm{d}^{-1}$ for snow and $0.0053\ \mathrm{mK}^{-1}\mathrm{d}^{-1}$ for ice as calibrated by Réveillet et al. (2017) for the Mer de Glace. The surface accumulation is the sum of the solid precipitation (snow) and winter liquid precipitation (rain); it is assumed that during winter any rain that falls freezes and remains in the snow pack. Previous works (e.g. Gerbaux et al., 2005; Réveillet et al., 2017; Vionnet et al., 2019) show that precipitation is underestimated by climate models. Comparison of precipitation simulated by Safran re-analysis (Durand et al., 2009) with the annual surface mass balance at Tacul between 1979 and 2015 and with the observed winter accumulation data available after 1994 in the accumulation area (see Supplementary Material) indicates that the Safran precipitation must be increased by 63% to best fit the observations, in good agreement with Réveillet et al. (2017). The same method is then repeated for the climate scenarios adopted for this study. For each scenario, the correction factor on precipitations is evaluated over the historical period 1979-2015. On the average, simulated precipitation must be increased by 70% to fit observations, with only slight differences from one scenario to another. The value of 70% is therefore applied to all scenarios. The surface mass balance at Tacul gate obtained after 2015 with the PDD model and the corrected precipitation from the 26 different climate scenarios constitute the forcing data for the 26 forecast simulations. The same relation as for the hindcast simulation (Eq. (7)) is then used to infer the spatial distribution of surface mass balance.

## 4.2 Flux through the Tacul gate

To account for the artificial boundaries at Tacul and Leschaux gates, normal ice velocities over these boundaries and changes in surface elevation are imposed as Dirichlet boundary conditions for the Stokes (Eqs. (3) and (4)) and free surface equations (Eq. (5)), respectively. The treatment is different for hindcast and forecast simulations, but also for Tacul and Leschaux, given that Leschaux has much less data.

In all cases, for the vertical evolution over the artificial boundary, we assume that the form of the vertical profile of the normal velocity is given by the Shallow Ice Approximation (SIA, Hutter, 1981). From the results of the inversion of basal friction performed over the whole domain using the 2003 observed surface velocity, we further assume a constant and uniform





ratio between sliding and surface velocities of $1/3$ at both gates ($r_{\text{slid}} = u_b/u_s = 1/3$). The vertical profile of the normal velocity at the gate is evaluated as

$$u(z) = r_{\text{slid}} u_s + u_d(z), \tag{8}$$

in which the deformational velocity is either imposed knowing $u_s$ (hindcast simulations at Tacul) as

$$u_d(z) = (u_s - u_b) \left( \frac{z - z_b}{H} \right)^{n+1}, \tag{9}$$

or evaluated using the diagnostic formulation for SIA (forecast at Tacul):

$$u_d(z) = 2A \left( \rho_i g \nabla H \right)^n (z - z_b)^{n+1}. \tag{10}$$

In the above equations, $z_b$ is the bedrock elevation whereas $H$ denotes ice thickness. In Eq. (10), the surface slope $\nabla H$ is the 2003 value and is held constant with time.

The transverse profile of surface velocity is assumed to follow the 2003 SPOT5 surface velocity at the Tacul cross section (see Fig. 7b in Berthier and Vincent, 2012): it is null on the side and increases linearly from both sides of the glacier to reach a maximum central value uniform over a constant width of 400 m.

For the hindcast simulation, this maximum central value of the velocity, denoted $u_{s\text{TAC}}$, is given from observations, as is also the case for the ice thickness $h_{s\text{TAC}}$. Knowing both the surface velocity $u_s = u_{s\text{TAC}}$ and ice thickness $H = h_{s\text{TAC}}$ in Eq.

(9), and assuming the above transverse velocity profile, the total flux through the gate can be estimated (see Fig. 2b). Despite the differences in the methods to estimate the ice flux at the gate, the inferred flux using this approach is consistent with the previous estimation of Berthier and Vincent (2012) (see Fig. 3) who assumed constant ratios of 0.8 between the width-averaged and observed center-line surface velocities and of 0.9 between depth-averaged and width-averaged surface velocities. The assumptions on transverse and vertical velocity profiles with $r_{\text{slid}} = 1/3$ we use in our modeling leads respectively to

ratios of 0.75 and 0.85, very close to the ones adopted by Berthier and Vincent (2012), explaining the closeness of the two approaches.

For the forecast simulations, $u_s$ and $H$ are unknown. Instead, the flux is directly evaluated from the integrated surface mass balance above Tacul gate (see Fig. 2b) and then used to determine the value of $H$ and the velocity distribution at the gate from Eq. (10). Ice flux through the gate is assessed by integrating, upstream of the Tacul gate, the surface mass balance given by

the climate scenarios. For steady state conditions, the ice flux should be equal to the sum of the surface mass balance obtained over the whole area of the upper part. In reality, the glacier being in a highly unsteady state, this condition is not fulfilled. To estimate the relationship between ice flux at the gate and surface mass balance upstream of the gate, we use the observations made between 1979 and 2015 and the reconstructed surface mass balance using Safran reanalyses. It is found that the observed ice flux at the Tacul gate is best estimated by averaging the surface mass balance integrated upstream of the gate over the 11

previous years (Fig. 3). It is furthermore assumed that this relationship will remain valid in the future.

The inferred relationships between ice flux, velocity and thickness at Tacul gate are shown in Fig.4. This figure also presents these relations for the available observations (1979-2015). Their comparison confirms the validity of the empirical relations



used above. As shown by Fig. 2b, some scenarios lead to a negative integrated surface mass balance above the Tacul gate, which could result in a very small or even null flux at the gate when integrated over 11 years. To avoid physically meaningless overly large decrease of $H$ (a zero flux would imply an instantaneous decrease of $H$ to zero), the annual decrease of $H$ at the Tacul gate is bounded by the local annual surface mass balance because the modeled thickness changes cannot be more negative than

ablation. Moreover, to ensure the physical consistency of this boundary condition over the whole simulation period, surface velocity and thickness cannot be null. In application of this second condition, the minimal thickness in our simulation is always greater than 70 m. For surface velocity, a minimal condition of $10 \, \mathrm{m \, a^{-1}}$ is applied.

The same protocol is repeated for the Leschaux boundary condition. Unfortunately, the ice flow velocities through the Leschaux gate are only available for the year 2003 from satellite data (Berthier and Vincent, 2012). For other years, we assume

that the ratio $u_{sTAC}(t)/u_{sTAC}(2003)$ obtained from Tacul observations is similar for the Leschaux gate. Note that in 2003, the surface velocity at the Leschaux gate is small ($9 \, \mathrm{m \, a^{-1}}$) compared to the velocity at the Tacul gate ($140 \, \mathrm{m \, a^{-1}}$). Its maximum ice thickness ($175 \, \mathrm{m}$) is half of that of Tacul gate ($360 \, \mathrm{m}$) while their widths are similar ($\approx 1000 \, \mathrm{m}$). The corresponding flux is consequently two orders of magnitude lower and its effect on the Mer de Glace flow is negligible during the period of interest. Therefore, for the forecast simulations, we simply assume that the thickness linearly decreases between the 2015 thickness and

a null thickness in 2050. The velocity profile is then directly given by Eq. (10) without estimating a flux from the upstream accumulation.

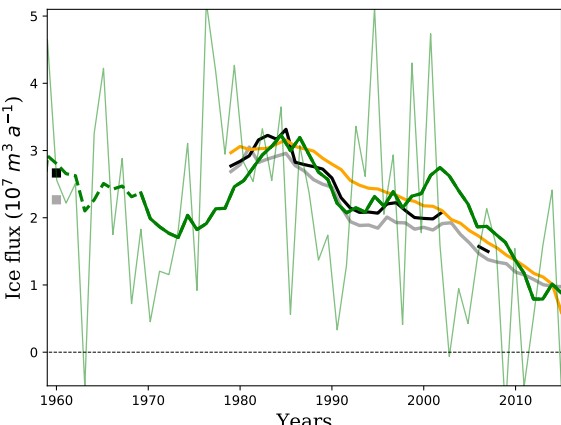

**Figure 3.** Ice flux through the Tacul gate from 1979 to 2015 based on a previous estimate (Berthier and Vincent, 2012, in black), from the SIA using only observed ice thicknesses at Tacul gate (orange), as imposed for the hindcast simulations (gray, see text) and compared to the yearly Safran surface mass balance integrated upstream of the Tacul gate (thin green) and its 11-year running mean (thick green).

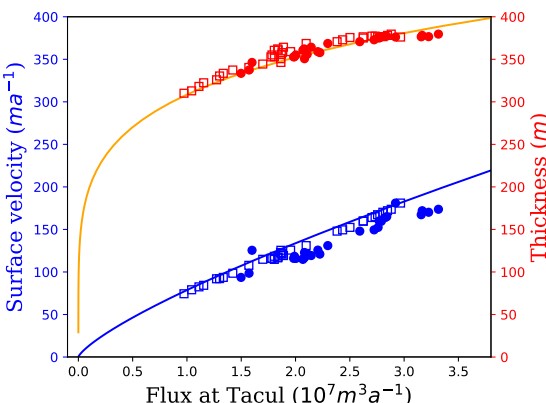

**Figure 4.** Surface velocity (blue) and thickness (red) at the Tacul gate as a function of the flux through the gate. The curves are the analytical solutions obtained using the SIA diagnostic formulation(Eq. (10)), the squares correspond to the flux integrated by Elmer/Ice using the observed surface velocity and ice thickness and a velocity distribution given by Eqs. (8) and (9). The circles are the fluxes estimated by Berthier and Vincent (2012).

## 5 Results

Figures 5, 6 and 7 show, respectively, the reconstructed surface velocity, elevation and front position for the whole period. Results from the hindcast simulation are compared to the observations over the period 1979-2015. After this validation stage, the forecast simulations explore the range of possible evolutions corresponding to the 26 EURO-CORDEX climate scenarios.

### 5.1 Hindcast simulation

For the validation of the hindcast simulation, the results of the model are compared with the observed surface elevation changes (Fig. 6) and centerline ice velocities at the four cross sections (Fig. 5). Note that for the highest observation profile (Tacul), the observations are used to impose the ice flux on this boundary of the model domain, explaining the perfect match between observations and model outputs. The validation is therefore only discussed for the three lowest profiles of Trélaporte, Echelets and Montenvers.

The overall good agreement of the model with the observations at the three lowest profiles was obtained without any tuning of the model parameters, except the inversion of the friction coefficient using the 2003 velocity and surface elevation dataset. The model is capable of reproducing the thickening phase in the first years of the simulation period with increasing ice velocity and ice thickness, as well as the subsequent thinning phase with decreasing surface elevation and velocity. Despite this good overall agreement, small differences are observed for both surface elevation and velocity.

For example, the peaks of calculated surface elevation and velocities are reached with a delay of about 3 years at Trélaporte. On the lower cross sections, Echelets and Montenvers, the surface elevation did not show a significant increase between 1979





and 1990. In general, the lower the profile, the larger the delay between the start of decrease of the simulated compared to the observed surface elevation. For all three of the lower cross sections, the modeled glacier is in general too thick over the last 25 years of the period compared to observations, with a maximum difference of up to $25\,\mathrm{m}$ for Montenvers, the lowest cross section. For this cross section, this overestimation decreases in the last years before 2015, eventually becoming an

underestimation. In general, the hindcast shows that the response time of thickness and velocities is too long, indicating that the modeled glacier does not respond quickly enough to the flux changes observed at the Tacul gate. The possible causes for this response delay are presented in the Discussion section.

Despite these local differences in surface elevation and velocity, the general trend of snout retreat is very well reproduced by the model over the whole hindcast period (Fig. 7). The simulated front is almost stable between 1979 and 1990 and starts

to retreat slowly 5 years before the observed retreat in 1995. Over the period 1995-2015, the observed rapid retreat of the ice front is well reproduced with a retreat rate of $30\,\mathrm{m\,a^{-1}}$ compared to $35\,\mathrm{m\,a^{-1}}$ for the observations.

## 5.2 Forecast simulations

The forecast simulations were carried out using the surface mass balance calculated from the 26 climate scenarios obtained in the framework of the EURO-CORDEX program (Fig. 2). Note that, whatever the representative concentration pathway

(RCP 2.6, 4.5 or 8.5), all these scenarios lead to a very similar mean decrease in surface mass balance until 2050 at Tacul gate (see Fig. 2a), with an almost doubling of ice lost in 2050 compared to 1960. As a direct consequence, the same trend is observed for integrated surface mass balance above Tacul gate (see Fig. 2b). Even if a few individual scenarios from all RCPs can lead to stable or even increasing integrated surface mass balance above Tacul gate until 2050, the general trend for all three RCPs is a decrease of surface mass balance, and therefore of the ice flux at Tacul gate that can drop to close to zero in 2050.

All forecast simulations show significant thinning and slowing downstream of the Tacul gate (Figs. 5 and 6). At Trélaporte and Echelets cross sections, the differences of thickness changes are within the range of $\pm20\,\mathrm{m}$ and $\pm10\,\mathrm{m}$, respectively, until 2030. Between 2020 and 2030, the thinning at Echelets profile is from 8.0 to $8.8\,\mathrm{m\,a^{-1}}$ (to be compared to the $5.0\,\mathrm{m\,a^{-1}}$ observed between 2005 and 2015). After 2030, the simulations show much larger differences induced only by the differences in surface mass balance obtained from the different climate scenarios. Note that each climate scenario influences both the ice

flux through the Tacul gate and the surface mass balance over the modeled domain. At the Tacul gate, depending on the climate scenario, the surface elevation could be either stable or could decrease by $250\,\mathrm{m}$ in 2050. For the most pessimistic climate model of RCP 8.5 scenario, the remaining ice thickness at Tacul gate is only $\approx 80\,\mathrm{m}$ in 2050, whereas the most optimistic scenario lead to a thickness slightly greater than that observed in 2015 ($330\,\mathrm{m}$).

However, these strong differences in ice thickness and ice flux at the Tacul gate lead to much smaller absolute differences in

thinning and ice flow velocity downstream of the gate; the lower the cross section, the smaller the response differences for the different scenarios. For instance, modeled thinning at Trélaporte only varies in a range of $2\,\mathrm{m\,a^{-1}}$ between 2030 and 2040, to be compared to differences as large as $9\,\mathrm{m\,a^{-1}}$ at the Tacul gate over the same period. Despite some scenarios indicating stable conditions at the Tacul gate, surface elevation and ice flow velocity at the three lowest profiles decrease until 2050 whatever the climate scenario, indicating highly unsteady state conditions for the present glacier.



Our modeling results make it possible to assess the retreat of the snout over the next decades (Fig. 7). The observed rate of retreat was $35\ \mathrm{m\,a^{-1}}$ for the hindcast period 1995-2015. According to the forecast simulations, the terminus of Mer de Glace glacier will retreat with rates varying from 60 to $85\ \mathrm{m\,a^{-1}}$ between 2020 and 2030, 65 to $95\ \mathrm{m\,a^{-1}}$ for the period 2030-2040 and more than $90\ \mathrm{m\,a^{-1}}$ after 2040. As a consequence, the Montenvers cross section could be free of ice by 2023

and the Echelets cross section by sometime between 2031 and 2035 depending on the climate scenario (Fig. 7). For the most pessimistic scenarios, the terminus could be close to the Tacul gate by 2050.

Finally, we define a mean reference scenario constructed as the average of all 26 climate scenarios. Figure 8 presents the evolution of the ice thickness and the glacier extent for this mean reference scenario. It also illustrates the variability in glacier extent induced by the different climate scenarios until 2050 by showing the minimum and maximum extent obtained with the

26 different scenarios at year 2015, 2025, 2040 and 2050. This mean reference scenario is further used to study the relative contribution of ice flux at Tacul gate and surface mass balance of the glacier tongue in the Discussion section.

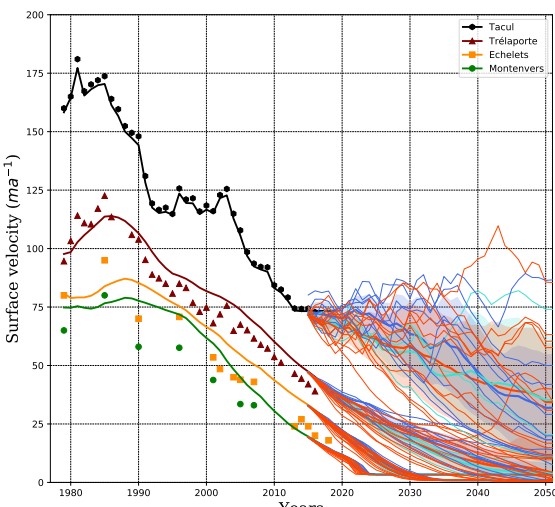

**Figure 5.** Surface velocity for all prognostic simulations. Hindcasts at the 4 profiles are shown by dark (Tacul), brown (Trélaporte), orange (Echelets) and green (Montenvers) curves and the symbols are the corresponding observations. Forecasts are shown in in dark blue (RCP 2.6), blue (RCP 4.5) and red (RCP 8.5), with the average forecasts represented by thick curves with $1\ \sigma$ uncertainty bands (colored area).

## 6   Discussion

The model reproduces the evolution of the glacier over the past four decades relatively well. However, the observed timing

and amplitude of changes are not perfectly reproduced and are increasingly inaccurate as the distance to the Tacul boundary condition increases. In particular, the modeled glacier is too thick and velocity too high, resulting in a flux that is increasingly

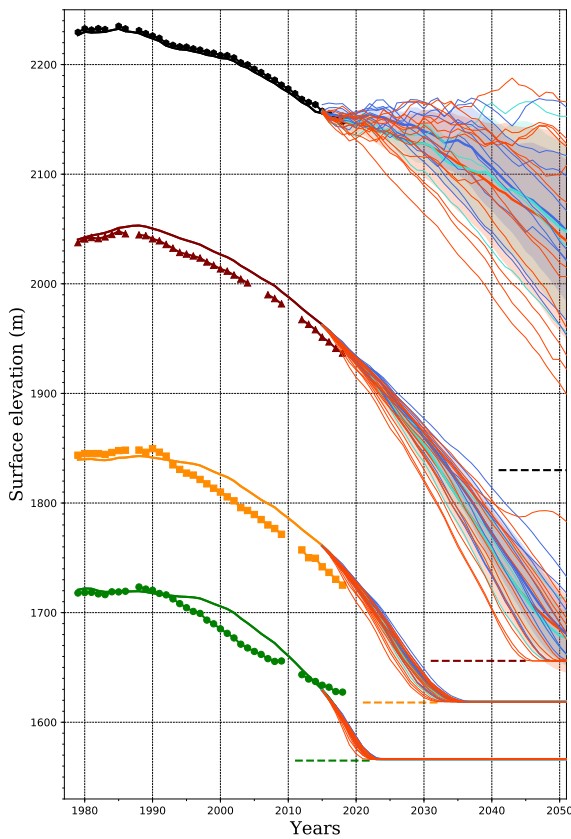

**Figure 6.** Same as Fig. 5, but for surface elevation. Dashed lines indicate the bedrock elevation for the 4 profiles.

too high at the profiles of Trélaporte, Echelets and Montenvers. For the hindcast period, there is a relatively high level of confidence in the applied surface mass balance and imposed flux at Tacul boundary condition, both being directly derived from a continuously maintained network of stakes over the whole glacier. According to Thibert et al. (2008), we can expect uncertainties for the ablation estimated from a network of stakes of the order of $0.14 \, \mathrm{ma}^{-1}$, which is low relative to the mean ablation measured on the tongue of Mer de Glace. Other uncertainties arise from the linear extrapolation of ablation over the tongue (Eq. (7)) based on measurements in an area of clean ice. Indeed, debris cover has increased in recent decades and may have locally decreased ablation by up to $3 \, \mathrm{m}$ (see Fig. 3b in Berthier and Vincent, 2012). This probably explains our overestimation of the thinning rate at the Montenvers profile after 2000.

The bedrock elevation, for which measurements have been greatly improved by a recent radar campaign over the modeled area (unpublished), is also well known with an estimated average uncertainty of $10 \, \mathrm{m}$ (Vincent et al., 2009). Moreover, velocities are not systematically over or under-estimated during the hindcast period, which might be an indication for a missing transient process in our simulation. Consequently, the major process explaining these discrepancies is likely basal friction and





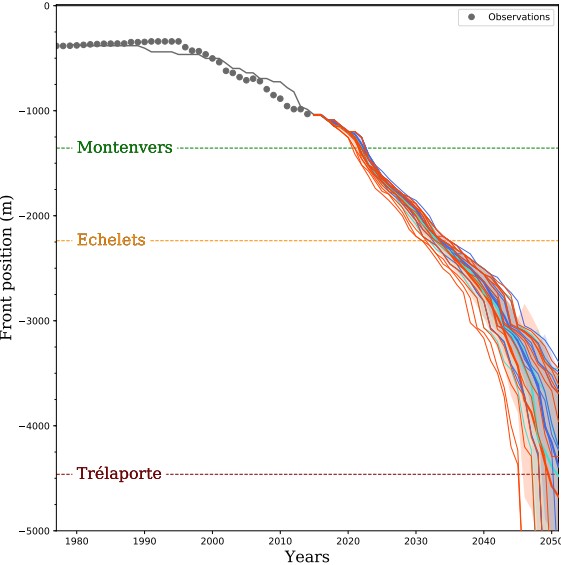

**Figure 7.** Evolution of the front position (along a flowline defined by the front fluctuation). The hindcast is in grey, and the squares represent observations. Forecasts are shown in dark blue (RCP 2.6), blue (RCP 4.5) and red (RCP 8.5) with 3 extreme scenarios underlined with thick curves and average forecasts represented by thick curves with 1 $\sigma$ uncertainty bands.

its evolution from year to year, not accounted for in the model. Indeed, the basal conditions are inverted from the 2003 dataset and kept constant over the whole simulation. Changes in glacier geometry and surface runoff have likely induced changes in basal conditions over the four decades. Inferring these changes would require contiguous surface DEMs and surface velocity maps, which are not available for dates other than 2003.

Regarding the glacier retreat, Berthier and Vincent (2012) estimated that over the period 1979-2008, two-thirds of the increase in the thinning rates observed in the lowest part of Mer de Glace was caused by reduced ice fluxes (and consequently emergence velocities) at Tacul gate and one third by rising surface ablation. In other words, they estimated that the retreat of the glacier front was more influenced by changes at high elevations than local changes. With a comprehensive ice flow description for the four last decades as well as for the future, the relative contribution to glacier retreat of local versus higher

elevation changes can be quantified. The results of our hindcast are consistent with the result of Berthier and Vincent (2012) over the period 1979-2008. For the future, we run a reference scenario that corresponds to the mean of the 26 scenarios (e.g. mean surface mass balance and mean flux at Tacul gate). The trends obtained for this reference scenario are general and apply to most of the other individual scenarios presented above. Contrary to the past trends, Fig. 9 clearly indicates that the future retreat of the glacier front will be influenced more by local changes (i.e. changes in surface mass balance over the lowest part

of the glacier) than by changes of flux from the upstream area (i.e. flux at Tacul gate). It is only after 2045, when the front approaches the Tacul gate, that its retreat starts to be largely influenced by changes in the upstream flux. The same trend is





visible for surface elevation changes at the two lowest profiles of Echelets and Montenvers where changes in surface elevation are mostly influenced by the local surface mass balance (Fig. 10). For the intermediate profile of Trélaporte, the influence from the flux at Tacul gate is visible, but the local surface mass balance still dominates the observed decrease of surface elevation over the whole studied period.

When surface mass balance is integrated over the whole glacier surface located downstream of the Tacul gate, we can estimate the relative contributions over time of the surface mass balance and the flux at the Tacul gate to the total change in volume of the glacier tongue. As depicted in Fig. 11, surface mass balance and flux at Tacul gate were almost equal (in absolute value) over the period 1979-1994 and the glacier tongue of the Mer de Glace was very close to equilibrium, explaining an almost stationary front position over this period (Fig. 7). Nevertheless, since that period, both terms have started to decrease, but not

at the same rate, explaining the two-third contribution of flux at Tacul gate observed by Berthier and Vincent (2012) over the period 1979-2008, and confirmed by our results. Whereas the flux at Tacul gate is decreasing at an almost constant rate since the mid-1980s, the rate of decrease of the tongue-integrated surface mass balance is evolving with time. As shown in Fig. 11, the tongue-integrated surface mass balance is currently reaching its minimum and will start to increase in the future. As a consequence, the volume lost from the tongue of Mer de Glace is currently reaching its maximum and will start to decrease in

the future. Indeed, even if larger melt rates are expected in the future, the tongue-integrated surface mass balance is increasing toward zero due to the decrease of the glacier tongue area. This explains why the surface mass balance over the glacier tongue is increasingly dominating changes in ice volume downstream of Tacul gate relative to the flux at the gate.

  Because of the adopted model domain, restricted to the part of the glacier downstream of the Tacul gate, it was not possible to conduct simulations after 2050 for most of the scenarios. Indeed, after this date, the ice thickness at Tacul rapidly becomes

null. The choice of adopting a restricted domain for the modeling was dictated by the lack of measurements of the bedrock elevation in the upstream part of the Mer de Glace. Prognostic simulations over a longer period would therefore require to acquire such data or to infer the bedrock topography using an inverse method (e.g. Fürst et al., 2018; Farinotti et al., 2019). Nevertheless, as shown by the sensitivity results, the evolution of the glacier tongue is not sensitive to this artificial boundary condition imposed at Tacul gate. Therefore, including the upper part of the glacier in the modeled domain would not likely

change the results over the studied period, but would allow further estimates after 2050.

## 7 Conclusions

In this study, the Elmer/Ice ice-flow model was applied to simulate the past and future evolution of the lower part of the Mer de Glace glacier. Given that the bedrock elevation remains unknown in the upper part of the glacier, we specified the ice fluxes at Tacul and Leschaux gates which are the upper limits of the tongue. These ice fluxes were obtained from the observed section

surface area and ice flow velocities in the past and assessed from the simulated surface mass balance in the accumulation zone for the future.

  The simulation of the tongue for the period 1979 to 2015 was driven by (i) surface mass balance measurements and (ii) the ice flux at Tacul and Leschaux gates that were obtained directly from the observed section surface area and ice flow velocities. Ice





flow modeling results were accurately compared to detailed and continuous observations of surface elevation, surface velocity and snout fluctuations over four decades during which the glacier both experienced a period of increase and a long period of decay. To our knowledge a comparison to data at this detail is unprecedented. We found that Elmer/Ice is able to reproduce the general behavior of the glacier. For example, the early growth of the glacier occurring between 1979 and 1990 is correctly

reconstructed. However, the elevation increase is underestimated in the lower part of the tongue. After 1990, the modeling results are in agreement with observations. We suspect that the small differences between the model and the observations could come from the assumed constant basal conditions over the hindcast period. Additional uncertainties on the surface mass balance of the tongue are likely related to the parse debris cover.

Using 26 climate scenarios, the model was run forward to simulate the evolution of the glacier tongue until 2050. There

were major differences in the ice fluxes calculated at the Tacul gate from all these scenarios, however changes in velocity and elevation at the lowest part of the glacier, as well as the retreat of the glacier front, were shown to be relatively independent of this upstream flux. Indeed, our sensitivity study indicated that the future changes at the lowest cross sections of the tongue are mainly influenced by the local surface mass balance, depending on the distance from the upper gate where the ice flux is assessed. This also explains why the upper cross section of Trélaporte is more sensitive to the upstream flux condition at

Tacul. Because of the decreasing surface area, the loss of ice volume of the part of the glacier downstream of the Tacul gate is currently reaching a maximum and will continue decreasing in the future. The glacier snout could retreat by 2 to 5 km over the next three decades and be close to Tacul gate in 2050.

Forecast simulations over a longer period would require extension of the model domain upstream of the Tacul gate, hindered by the unknown bedrock topography. Radar measurements in the upper part of the Mer de Glace and/or inverse modeling are

therefore required to estimate the bedrock topography in this area before realistic forecast simulations of the Mer de Glace can be extended beyond 2050.

### Acknowledgement

We thank Samuel Morin and Deborah Verfaillie (Centre d'Etude de la Neige, Météo France/CNRS) who produced the climate scenarios used in this study (ADAMONT simulations, https://opensource.umr-cnrm.fr/projects/adamont). Remote field mea-

surements were obtained thanks to the support of the TOSCA program of the French Space Agency (CNES). Glaciological data were obtained from the GLACIOCLIM database, https://glacioclim.osug.fr/. This study was supported by Electricité De France and Compagnie du Mont Blanc funding. We thank all collaborators past and present for their thorough measurements of mass balance, thickness and velocity changes as well as bedrock topography on Mer de Glace over all these decades.



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





**Figure 8.** Ice thickness and glacier extension a) at the end of the hindcast simulation in 2015 and for the mean reference forecast simulation in year b) 2025, c) 2040 and d) 2050. The climate scenario for the mean reference forecast simulation is the average of all 26 climate scenarios. Extensions of the most optimistic and most pessimistic scenarios are plotted in dark blue and red, respectively. The initial glacier extension in 2015 is plotted in black. The background image is the orthophotoplan from 2008 (©RGD74).





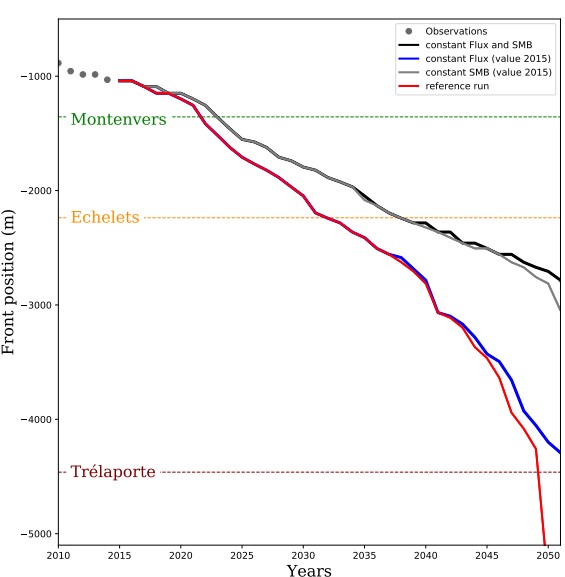

**Figure 9.** Sensitivity experiment for the mean reference scenario assuming the mean surface mass balance of all scenarios. Evolution of the glacier front for this mean reference scenarios (red), assuming a constant surface mass balance (grey, value from year 2015), assuming a constant flux at Tacul gate (blue, value from year 2015) and assuming that both surface mass balance and flux at Tacul gate are constant and equal to their 2015 values.



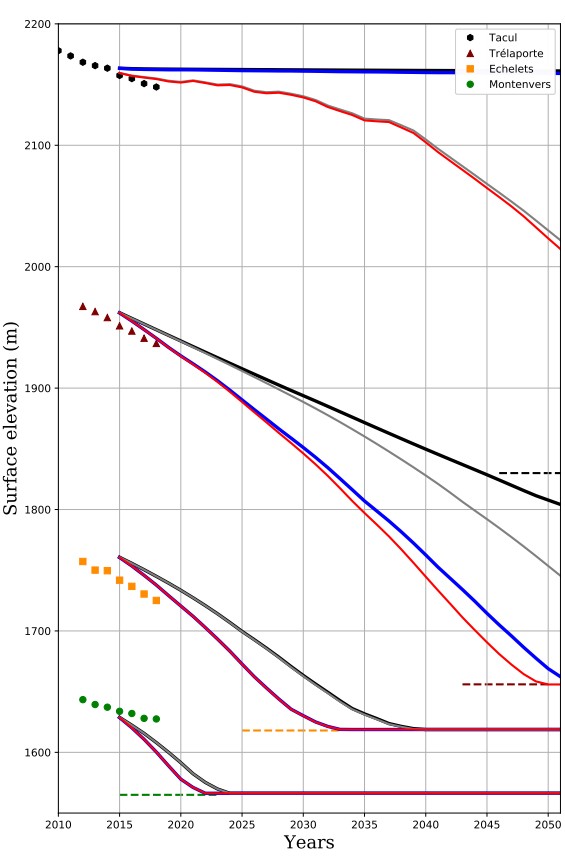

**Figure 10.** Same as Fig. 9 but for the surface elevation. Dashed lines indicate the bedrock elevation for the 4 profiles. For the two lowest profiles of Echelets and Montenvers, the red and blue curves are superimposed.



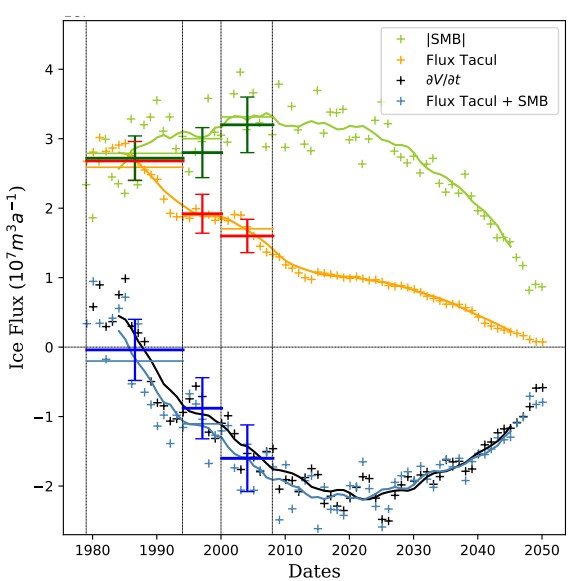

**Figure 11.** Sensitivity experiment for the mean reference scenario assuming the mean surface mass balance of all scenarios. Evolution with time of the absolute value of the integrated surface mass balance (green, real value always negative), integrated flux at Tacul gate (orange, always positive) and changes of volume of the glacier tongue (black) in cubic meters of ice per year. The blue curve represents the sum of the two fluxes and is almost equal to the change in volume. For each quantity, crosses represent annual values whereas the curve is a 10-year running average. The bars with error bars in dark colors are the estimates of the same quantities by Berthier and Vincent (2012) for the 3 periods delimited by the vertical gray lines. Horizontal lines using the same colors as the curves represent the averages of the different quantities over the same periods.