# Peer review of "Numerical modeling of the dynamics of Mer de Glace glacier, French Alps: comparison with past observations and forecasting of near future evolution."

_The Cryosphere, 2020_

## Referee Comment (RC1) · Anonymous Referee #1 · 5 May 2020

**General comments**

Mer de Glace is the largest french glacier, and probably one of the best known to the general public in France. In the context of climate warming, its future becoming is therefore of high societal interest, especially because it is a major tourist attraction of the Mont-Blanc area.

[Figure]

In this paper, the authors address the question of the future evolution of the Mer de Glace using a state-of-the-art ice flow model to simulate the 3 next decades considering a large panel of climate scenarios. The model was calibrated using a large set of data (including mass balance measurement, bedrock information, surface ice flow observation among others) and validated over the last decades against historical records. As a key result, a strong retreat of the Mer de Glace – from 2 to 5 km – must be expected within the next 3 decades.

To my knowledge this is the most advanced model of the Mer de Glace. The new results permit to refine of an earlier prognostic given by Vincent and al. (2014), who used a model of lower complexity. I am very confident it will not only interest the community of glaciologists, but also skateholder of the economic life (e.g. tourism, hydro-power production).

My feeling is that the manuscript is already in a good shape. I don't have any major concerns about the methodology, which is well established. Also the interpretation of results makes sense to me. I think two aspects of the manuscript still have room for moderate improvements (check the details in the main comments below): i) comparison with other studies treating the future of glaciers in the European Alps to "contextualize" your results ii) the presentation of the paper (structuring and English language). I also reported below more specific and technical comments, which I hope will help to improve the paper before it gets published.
**Main comments**

- Section 3 on the ice flow modelling contains a lot of content, which is standard in the Elmer Ice workflow and widely used. I don't see the point of recalling this again (in particular the set of equations) in the body of the paper. I suggest to replace it by appropriate references, or to move it in appendix or supplementary material to keep the paper concise and focused. Only what is specific to the Mer de Glace case should be kept in the paper's body (e.g. the choice of $A$ relative temperate ice, the implementation of given fluxes on the domain borders, ...) Details about the 3D mesh should also be moved to appendix. Last, modelling details normally belong to 'Methods', so I suggest to move a very concise version of Section 3 (Ice flow) to Section 4 (Methods), and possibly give further details in appendix.

- While you mention similar existing Full Stokes-based models, or global glacier models of the Alps in introduction, you make no link at all in the discussion. Typically, questions like "How do your results compared with some recent global modelling by Zekollari and al. or other FS model of individual large-scale glacier such as Aletsch?" or "Is the Mer de Glace expected to resist better than other neighbouring glaciers?" could be addressed in your paper. Comparisons with recent other models, and other glaciers of the Alps would put your results in a broader context. In abstract and conclusion, you write "To our knowledge a comparison to data at this detail is unprecedented": This statement is vague and unsupported in the paper.

- I feel the reading of the paper can be made more efficient if the English was improved by a professional. Line 10-12 p6 is an example of phrasing with repetitions that can be improved.

**Specific comments**

- I do not fully understand your choice of mass balance approach for the forecast simulation. A logical choice would be to simply compute the PDD everywhere instead of doing so at the Tacul gate only, and extrapolating the MB via Eq. (7). PDD can computed everywhere assuming vertical lapse rate for temperature. By doing so, you would release your hypothesis of linearly changing mass balance with respect to $z$. Don't you lose any information by this assumption? You could possibly tune PDDs to fit observation at Tacul in the past.

Eq (7) Something must be wrong with this equation since $b = 0$ when $z_s = z_{sTAC}$ while it should be $b = b_{TAC}$, isn't it? Also, from $k_b = \frac{\partial b}{\partial z}$ one expects $k_b$ to be dimensionless. From Eq. 7, one expects instead $k_b[z_s - z_{sTAC}$ to be dimensionless. Please double check. Also you have not defined $z_s$ and $z_s = z_{sTAC}$.

l27-30 p8 'relationship between ice flux at the gate and surface mass balance upstream of the gate': Could you elaborate your method (and possible its limitations) based on the two following questions: i) can your result of 11 years be interpreted as the average time for the ice to travel from the surface to the gate? ii) This '11 year' result is probably highly dependent of the glacier regime (advancing, retreating, highly retreating ...), which is fine if you modelled the retreat over the next 3 decades, but if this is the case, can you elaborate? and thus equivalently justify your sentence l30 : "It is furthermore assumed that this relationship will remain valid in the future."

l16 p12 'the modeled glacier is too thick and velocity too high': this surprises me, I would expect that it is too thick because too slow following simple mass conservation principle. How do you explain that ice can be both too thick and too rapid?

- I missed where you describe the 'sensitivity experiments' of Fig. 9 and 10 in your text? I understand that the goal is test the influence of the artificial flux condition at Tacul, which is important as you only model part of the glacier by lack of data.

However, this should be explained in the body of the paper, and not being left alone in Fig. 9 and 10.

**Technical comments**

Abstract I suggest to start straight the abstract with the 'Mer de Glace' and remove general statements (i.e. the 3 first sentences) to make the abstract more focused.

l 6-7 p1 'Mer de Glace (Mont Blanc area)' ⇒ 'Mer de Glace glacier, France'.

Abstract As said earlier, 'To our knowledge a comparison to data at this detail is unprecedented.' is (too?) vague statement and not supported.

Abstract It is hard to evaluate how much represent 2 to 5 km without having any prior idea of the glacier size, could you instead use (or add) a relative metric? i.e. 30% to 65% of the glacier length.

l 21 p1 Suggest changing 'fluctuations' into 'evolutions'.

l 4 p2 'Process-based model were also developed to take into account simple dynamics (...)' gives the feeling that first physically-based ice flow models started in 2015, which is obviously not the case. Could you reference early SIA-based models applied to mountain glaciers to fix this?

l 8-14 p2 You should clearly write here that by 3D physical model, you mean Stokes without any shallow-like approximations that would reduce the dimension of the equations (unless I'm mistaken?).

l 10-14 p2 Can you reduce the number of citations? Do you need all of them? It is anyway not exhaustive.

l 19-21 p2 'In addition, running simulations ... over three decades.' This is crucial here to justify the use of 3D Full Stokes to capture the complexity of the Mer de Glace ice flow, however, this is not exactly what this sentence is doing. I suggest to reformulate it to emphasize that using such a state-of-the-art model is not just an additional opportunity, but necessary to model the complex ice flow. Otherwise the reader might wonder why a simple model can not do the job.

l 21 p2 'This dynamics make it necessary to take into consideration the delay in the glacier response to climatic forcing.' This justifies to take into account the ice dynamics, but not to use a sophisticated model. Just make sure the argument is used for the right reason.

l 31-32 As shown in Fig. 1, Mer de Glace glacier has a single tributary glacier, which is named Leschaux glacier.

l5 p3 'maintained' ⇒ 'that are monitored'

l6 p3 I expected to have the elevations given.

l8 p3 ... they **were** not ...

l11 p3 'paucity' ⇒ 'lack'

l13 p3 .. the model domain **was** restricted

l14-15 p3 A transition causality word to connect the two last sentences is missing.

l19-20 p3 Again can you reduce the number of citations?

l29 p3 Rheological parameter with a constant value assuming temperate ice (A = 5.0159 $e^{-24}$), do you mean $10^{-24}$ ? Could you give a reference for this value? Possibly consider change the unit into 158 MPa$^{-3}$ a$^{-1}$. This is twice the value of Cuffey and Paterson (2010).

| l 8-9 p6 | **The two next sections describe** in detail the respective boundary conditions for the two steps (hindcast and forecast) defined at the surface **(mass balance)** and at the Tacul and Leschaux gates (ice flux from the accumulation areas), **respectively.** |

l10-12 p6   4 times 'surface mass balance' in 3 lines, can you improve the text? (or use an acronym)

l 8 p7   Define RCP and shortly comment of the meaning of the 3 pathways.

l10 p7   'good performance', this statement calls for ad hoc references.

l17 p7   'is underestimated by climate models': Do you mean in general or for the climate model you specifically use?

l21-22 p 7   For each scenario, the correction factor **for precipitation** is evaluated over the **past** period 1979-2015.

l 38 p10   'Fig.4.' $\Rightarrow$ 'Fig. 4.'

l6 p8   'using the diagnostic formulation for SIA' $\Rightarrow$ 'using the SIA'. Why qualifying the SIA formula as diagnostic?

(9) and (10)   Implementing Eq (10) straight without any correction sounds hazardous as the chance to get unrealistic velocities is high as the SIA model is local (sensitivity w.r.t $\nabla s$) and not tuned. In contrast, adding a correction factor that multiply Eq. (10) and is tuned to ensure continuity in time between Eq. (9) and (10) would make sense.

Eq. (10)   As you know the thickness here, did you consider implementing the SIA with a "shape factor" that accounts for lateral resistance of the ice flow as function of the type of profile?

l14 p8 'his maximum central value of the velocity', do you mean 'maximum value along the transect'?

l15 p8 'above transverse velocity profile': refer to Eq. (9).

l16 p8 'Despite the differences in the methods' I'm not sure what are the 2 methods you referring: Eq. (9) vs Eq. (10) ? Please, clarify.

l28 p8 Repeat the reference to Durand for the Safran reanalysis.

l15 p11 It can be surprising to see so tiny differences between different RCPs. You may say that this was expected as RCPs differ substantially more after 2050 (if I'm not mistaken).

l 30 p11 "However, these strong differences in ice thickness and ice flux at the Tacul gate lead to much smaller absolute differences in thinning and ice flow velocity downstream of the gate; the lower the cross section, the smaller the response differences for the different scenarios." If you were normalizing the model discrepancies by the ice thickness or ice velocity at the gate, the differences would probably be more uniform, isn't it?

l1 p12 'assess' ⇒ 'estimate' or 'forecast' or 'predict'

l5 p13 'which is low relative to the mean ablation measured on the tongue of Mer de Glace' can you give a number?

l6 p13 'Other uncertainties arise from the linear extrapolation of ablation over the tongue (Eq. (7)) based on measurements in an area of clean ice', your assumption of linear MB w.r.t altitude might by the highest source of uncertainty? If yes, it would make sense to start with this.

l8 p13 Please reference to a Figure.

l10 p 13 'is also well known' $\Rightarrow$ 'is also well constrained'

l11 p 13 not underestimated **by the model** ... But Fig. 5 shows a slight but real systematic over-estimation of velocities, isn't?

l12 p 13 To what 'discrepancies' do you refer? In the sentence before, you say there is no major discrepancies in velocities.

l1-4 p14 One can also argue an inaccurate sliding ratio to explain model errors? Have you considered testing different rate factors $A$ to increase or decrease vertical deformation?

l13 p15 'will start to increase ...' $\Rightarrow$ 'is expected to increase ...', check not to use future tense when you describe model results, but conditional or equivalent. Same for the sentence right after.

Fig 2. smb not defined. How is it possible that different RCPs show different values in the past times?

Fig 4. formulation(Eq. (10)),

Fig 4. observed surface velocity**,** ice thickness**,** and a velocity distribution given

Fig. 6 "Same as Fig. 5, but for surface elevation." Please make a full sentences. For convenience, better to repeat the text to make each individual caption self-contained. An other possibility is to merge Fig. 5 and Fig. 6, with two panels. The same for Figs. 9 and 10.

Fig. 8 colorbar is missing

Fig. 10 Color coding missing. Again the Figure would be better if self-contained without having to look for the color coding in another Figure. Otherwise merge Fig. 9 and 10 with two panels.

[Figure]

---

## Referee Comment (RC2) · Anonymous Referee #2 · 31 May 2020

[referee-annotated manuscript omitted]

---

## Referee Comment (RC3) · Anonymous Referee #3 · 2 Jun 2020

**Review of 'Numerical modeling of the dynamics of Mer de Glace glacier, French Alps: comparison with past observations and forecasting of near future evolution'**
by Vincent Peyaud et al.
The Cryosphere Discussions – May 2020

In this manuscript, Peyaud and colleagues use the Elmer/Ice ice flow model to simulate the dynamics and the evolution of the Mer de Glace between 1979 and 2050. They pay elaborate attention to the evaluation of the model output by comparing their simulations to an elaborate dataset collected on the glacier between 1979 and 2015. Subsequently, they model the future evolution of the glacier under 26 different climate scenarios, for which they particularly focus on the modelled changes in glacier length and velocity.

This is a nice and detailed study, which has been well designed and is pleasant to read. The manuscript will be of interest to glacier modellers and fits very well in *The Cryosphere*. The level of detail of ground-truth data is very high and it is nice to see such an elaborate effort to compare these with modelled results. The list of comments and suggestions provided below may seem relatively long at first, but most of the comments should be relatively easy address and to incorporate.

**General comments**

- It is a pity to see that only the lower part of the Mer de Glace was modelled (as explained in section 3.2 and 4.2). While going through the manuscript, I was constantly thinking: why is this the case? The explanation, in which this is linked to the uncertainty in the bedrock, appears only towards the end of the manuscript (p.15, l.20-21). If you were to consider the upper part of the glacier also, you would indeed introduce additional uncertainty in your simulations; but you now also do so by imposing several conditions on the fluxes through the upper gates (Tacul and Leschaux): e.g. linking the flux at the gates with upstream integrated surface mass balance based on observations and imposing this for the future (while in reality the glacier response time will play a big role here). It would really be nice to see how much this influences your results by having some additional sensitivity tests in which you modify the imposed model settings. Even better would be to have some simulations in which you model the entire glacier (i.e. after inverting the ice thickness in the accumulation area) and see how they compare to your results. Would this be feasible? Such a test would require some additional work, but I honestly think that this would add a lot to your story and would also increase the impact of your story (as it could be used as a kind of reference for future studies that impose fluxes at gates and only model a part of the glacier – a method that may definitely gain in popularity for certain applications!)
- The level of detail of your analyses is very sophisticated and you consider several elements in your evaluation and for the projections that many other studies do not include.
  - However, it would also make sense to have insights in the more widely considered glacier characteristics, such as glacier volume and area:
    - For past: with this you can directly assess the performance of your model to reproduce e.g.:
      - Past volume changes (would in fact be a kind of test for your surface mass balance model in this case), which can be derived from DEM differencing (and which I imagine is maybe already directly available for this glacier?).
      - Past area changes.
    - For future: allows you to compare more easily to other studies in which the evolution of Mer de Glace is also modelled and comparison with other glaciers in the European Alps (e.g. is Mer de Glace more/less retreating than other glaciers,...; list of studies is provided further on)
  - In general, you refer to your study as an elaborate evaluation, which it definitely is, and which I think is very impressive. However, some of the agreement may also results from several choices you made, which are not always explained (see comment below on this). It is therefore difficult to disentangle which part of the agreement results from a kind of calibration ('tuning') and is therefore not a real kind of validation/evaluation (as you want the calibration and the evaluation data to be – ideally entirely – independent).
- There is a sort of discrepancy between the complexity of the model used for the ice dynamics and the relatively rough approach for surface mass balance and imposed boundary conditions at the gates.

Given that the glacier is so well studied, why did the authors not consider more complex approaches for this (e.g. thinking of e.g. debris cover; constant mass balance gradient)? A few additional sentences and motivation would be nice.

- Some assumptions are made, and it is not always clear how these affect your results. Would be good to have some additional insights in the sensitivity of your findings to your various assumptions. This includes assumptions related to:
    - Constant mass balance gradient
    - Imposed ratio between sliding and surface velocities at the Tacul gate
    - Assumption that relationship between ice flux at Tacul gate and integrated surface mass balance for upstream area remains the same
    - Minimal thickness and velocities at the gates
    - Linear decrease ice thickness at Leschaux gate over time,
    - ....

    For a full list and more details, refer to the specific comments below.
- Most of the figures could be improved relatively easily to enhance their readability: see suggestions below.

**Specific comments and suggestions**

Abstract:
- p.1, l.2: 'All alpine glaciers are shrinking and retreating at an accelerating rate...': technically this is not entirely true. It is the case for most glaciers, but there are some exceptions (e.g. glaciers that are almost gone or those that disappeared; i.e. where the retreat does not accalerate). Suggest changing this to: 'Alpine glaciers are shrinking and rapidly lose mass in a warming climate'
- p.1, l.8-9: 'To our knowledge a comparison to data at this detail is unprecedented': indeed a very detailed comparison to data is present, which is very nice. But not sure you can claim that it is unprecedented, as comparisons to other studies are not straightforward (in some studies other types of data have been considered). Probably best to remove from abstract and mention in this in the main text, where there is room for more nuance. Check studies on individual glaciers with elaborate evaluation and/or calibration with ground-truth data (e.g. Adalgeirsdóttir et al., 2011; Zekollari et al., 2014; Hannesdóttir et al., 2015).
- p.1, l.9-10: You mention the velocities and the elevation changes for the model evaluation. What about the mass balances and the length variation, which you mention a few sentences before (in l. 6): how do these perform? This becomes clear in the text, but for consistency would be good if you could already mention them here.

1 Introduction:
- p.1, l.19: 'sea-level rise': could be worth referring to the new GlacierMIP studies, in which the future sea level contribution from glaciers are obtained from a community-wide intercomparison effort (Hock et al., 2019; Marzeion et al., 2020)
- p.1, l.21-22: 'first studies': you are not very specific here. Given that you model a single glacier and have not mentioned the 'large-scale' glacier modelling aspect yet, one would assume that these are the first studies for the evolution of individual glaciers in the European Alps. I suggest being more specific here (mentioning the regional aspect) and/or to refer to pioneering studies in which ice dynamics are included for individual glaciers (e.g. Huybrechts et al., 1989; Letréguilly & Reynaud, 1989; Stroeven et al., 1989; Greuell, 1992). Would somehow be strange to spend your introduction focusing on large-scale glacier modelling, while your work in fact focuses on very detailed glacier modelling.
- p.2, l.6: better also update with the new numbers from the second GlacierMIP effort (Marzeion et al., 2020)
- p.2, l.10-12: list of references for 'model describing the complex three-dimensional geometry of a whole glacier' is a bit odd:
    - Some studies do not take into account the glacier evolution over time
    - Others are in fact based on the SIA , which makes them rather 2D (as described in the title of Le Meur & Vincent, 2003) and more in line with what you describe earlier (p.2, l.4-5) as 'Process-based model ... to take into account simple dynamics' (Clarke et al., 2015)

- o The ITMIX experiment, which focuses on ice thickness reconstruction (Farinotti et al., 2017), is also a bit odd to mention here
- o Why not simply focus on what you also do here: 3-D time-evolving simulation of a single glacier? (e.g. Schneeberger et al., 2001; Le Meur et al., 2004; Jouvet et al., 2009, 2011; Zekollari et al., 2014; Ziemen et al., 2016; Jouvet & Huss, 2019; Gilbert et al., 2020; Schmidt et al., 2020). Would also be interesting for the discussion to compare your modelled future evolution of Mer de Glace with the modelled evolution of other glaciers in the European Alps (see also general comment on this).
- p.2, l.21: 'This dynamics' → 'These dynamics'

3 Ice flow model:
- p.3, l.29: Value for the rheological parameter for ice: how was this value chosen? Quite often this is used as a calibration parameter as it has a large influence on the ice thickness (/glacier volume). By just taking a value from literature: difficult to assume that this will work well immediately for your glacier of interest. See studies in which this was analysed / where this rheological parameter was tuned (e.g. Schmeits & Oerlemans, 1997; Albrecht et al., 2000; Vincent et al., 2000; Giesen & Oerlemans, 2010; Adalgeirsdóttir et al., 2011). From my understanding, in your study the calibration occurs through the basal sliding, in which you try to match observed velocities: but what is effect of this approach on modelled ice thickness evolution? i.e. How are you sure that the modelled evolution is related to physical forcing and not to some kind of model drift? Would be good if you could explain this a bit in the manuscript.
- p.4, Figure 1: would have been nice to have surface elevation information in this figure (vs. visual imagery). Through this, would be easier to orient for someone who's not very familiar with the glacier.
- p.5, l.16: model domain does not cover the entire glacier. Why? (I saw later that this is explained towards the end of the manuscript) Should really clarify this choice. Pity to not have the entire glacier in / or additional experiments in which this is the case to compare to,
- p.5, l.28-29: 'Bedrock elevation...interpolation (Fan et al., 2005) of all available observations': does this mean that the bedrock is simply obtained from a kind of kriging? Is it not justified to rely on a more sophisticated approach, especially given the fact that you then use a very complex 3-D model to solve for dynamics and temporal evolution? Would also be good to have an idea where the ice thickness (/bedrock elevation) was measured (unpublished data is mentioned later; but maybe you can add the profiles in a figure somewhere?).

4 Methods:
- Name of the section ('Methods') is maybe not ideal, as in fact the previous section ('Ice flow model') is also really part of the methods. As you mainly describe the boundary conditions here (at the cross sections), you could consider renaming this section 'Boundary conditions' or something alike?
- p.6, Figure 2:
  - o Quite difficult to decipher this figure: the grey line, which represents the 'average' is barely visible in the right panel.
  - o Not ideal to combine green and red colours for lines in a single figure, given that a considerable amount of people cannot see the difference between these two colours (see e.g. https://en.wikipedia.org/wiki/Color_blindness#Red%E2%80%93green_color_blindness)
  - o One needs to look up in the caption what the average stands for, maybe specify that this is the average of the RCPs? Same of Safran: maybe good to specify this, as not clear what this is at this point in the manuscript (i.e. Reanalysis 'SAFRAN')
  - o Do not entirely get why you show RCP's for the past and how this should be interpreted. Makes sense that these are off if they have not been forced with reanalyses product (e.g. ERA5). With this, expect them to be much closer to SAFRAN reanalyses product also. Also, not entirely clear if what you show here is the SAFRAN original SMB, or the one that is corrected by scaling the precipitation with ca. +60-70% (as you describe towards the end of section 4.1.). I expect the latter, given the good agreement in SMB. If so, and if I understand it correctly, would it also make sense to have the 'modified' SMB (with precipitation correction) from the RCPs?
- p.6, l.5-7: 'For the forecast simulations from 2015 till 2050, results from climate simulations are used to evaluate the flux on the different boundary conditions of the glacier domain': I get the meaning of this sentence, but it is a bit strange / misleading to use 'evaluate' here, as this is what you use to

describe the evaluation of the hindcast also. Maybe change to: '…are used to simulate the future flux evolution at the boundary of the glacier domain'

- p.6, l.6: 'till' → 'until'
- p.7, l.4-5: 'Despite this strong variability from year to year (Fig. S1 in the Supplementary Material and Rabatel et al., 2005), a constant mass balance gradient of kb = 0.009 m$^{-1}$ is adopted for hindcast and forecast simulations'. How does this affect your results / how would your result look like if you take into account the interannual variability and also not rely on a constant gradient?
- p.7, l.6-9: why did you consider these 26 future climate projections from the EURO-CORDEX ensemble, given that there's many more (>50) available? Any criterion used to choose only those? Moreover, are these simulations from the RCM at 0.11° resolution or at 0.44° resolution (or a mix?). Would be good if you could be a bit more specific on this.
- p.7, l.34 – p.8, l.1: 'we further assume a constant and uniform ratio between sliding and surface velocities of 1/3 at both gates': what is this assumption based upon? Given the lack of direct observations of basal velocities, the uncertainty on this statement is quite large. How does this influence your results?
- p. 8, l. 28-30: assumption about relationship flux at Tacul glacier and integrated surface mass balance higher area. You assume that this remains constant in the future: how much does this influence your results?
- p.9, l.7: imposed values for minimal ice thickness and surface velocities. Why are these values chosen? And what is influence on your simulations?
- p.9, l.14-15: linear decrease in ice thickness at Leschaux gate: again, sounds rather arbitrary. What is effect of this on your simulations? Can imagine that this could have quite an impact on projected future changes..
- p.9, Figure 3 + p.10, Figure 4:
  - Like for figure 2: would make sense to have information on what the different lines represent in the figure itself instead of in the caption. There is enough space, and would make the interpretation much easier and more intuitive (and an advantage if you plan on using this in a presentation later!)

Results:
- Although you do not really calibrate to this, it is not fully clear how the several choices you have made affected your simulations and can therefore be considered as an independent evaluation.
- p.11, l.1-2: 'In general, the lower the profile, the larger the delay between the start of decrease of the simulated compared to the observed surface elevation': this does not really come as a surprise, as you impose the fluxes at the Tacul gates to fit observations. As such the uncertainty in your results 'spreads' as you go away from these points (towards lower glacier elevations).
- p.12, Forecast simulation: part of the similarity in future evolution is driven by the fact that the SMB is quite similar, as the difference in the forcing (temperature and precipitation) increases with time and in fact only becomes really notable during the second part of the 21$^{st}$ century. However, a part of this is also simply related to the glacier response time, which is in the order of decades for this glacier. Would probably be worth placing this in context a bit (potentially in the discussion session) and making the link to response time studies on Alpine glaciers (e.g. Zekollari et al., 2020).
- Fig. 5-7: would again be good if could read the figure without having to refer to the legend (i.e. add information about the RCPs in figure directly). For RCP colours, it would be nice if you could use more conventional colours for the different RCPs (e.g. those used in the IPCC reports). Finally, mixing green and red in the same figure is still not a very good idea.

Discussion & conclusion:
- Nicely elaborated and very interesting in general!
- p.13, l.6-8: role of debris is mentioned. Given the complexity of your ice flow model and the level of detail of your analysis, would it not make sense to incorporate debris cover in your approach (and eventual evolution over time; see e.g. Jouvet et al., 2011)? Or maybe do some tests in which this is incorporated in a parameterized way to analyse whether this decreases the discrepancy between observations and modelling that you mention here.

**References**

Adalgeirsdóttir, G., Gudmundsson, S., Björnsson, H., Pálsson, F., Jóhannesson, T., Hannesdóttir, H., et al. (2011). Modelling the 20th and 21st century evolution of Hoffellsjökull glacier, SE-Vatnajökull, Iceland. *The Cryosphere*, *5*(4), 1961–1975. https://doi.org/10.5194/tc-5-961-2011

Albrecht, O., Jansson, P., & Blatter, H. (2000). Modelling glacier response to measured mass-balance forcing. *Annals of Glaciology*, *31*, 91–96.

Clarke, G. K. C., Jarosch, A. H., Anslow, F. S., Radić, V., & Menounos, B. (2015). Projected deglaciation of western Canada in the twenty-first century. *Nature Geoscience*, *8*, 372–377. https://doi.org/10.1038/ngeo2407

Farinotti, D., Brinkerhoff, D. J., Clarke, G. K. C., Fürst, J. J., Frey, H., Gantayat, P., et al. (2017). How accurate are estimates of glacier ice thickness? Results from ITMIX, the Ice Thickness Models Intercomparison eXperiment. *The Cryosphere*, *11*, 949–970. https://doi.org/10.5194/tc-11-949-2017

Giesen, R. H., & Oerlemans, J. (2010). Response of the ice cap Hardangerjøkulen in southern Norway to the 20th and 21st century climates. *The Cryosphere*, *4*(2), 191–213. https://doi.org/10.5194/tc-4-191-2010

Gilbert, A., Sinisalo, A., Gurung, T., Fujita, K., Maharjan, S., Sherpa, T., & Fukuda, T. (2020). The influence of water percolation through crevasses on the thermal regime of a Himalayan mountain glacier. *The Cryosphere*, *14*, 1273–1288.

Greuell, W. (1992). Hintereisferner, Austria: mass-balance reconstruction and numerical modelling of the historical length variations. *Journal of Glaciology*, *38*(129), 233–244.

Hannesdóttir, H., Aoalgeirsdóttir, G., Jóhannesson, T., Guomundsson, S., Crochet, P., Ágústsson, H., et al. (2015). Downscaled precipitation applied in modelling of mass balance and the evolution of southeast Vatnajökull, Iceland. *Journal of Glaciology*, *61*(228), 799–813. https://doi.org/10.3189/2015JoG15J024

Hock, R., Bliss, A., Marzeion, B., Giesen, R. H., Hirabayashi, Y., Huss, M., et al. (2019). GlacierMIP – A model intercomparison of global-scale glacier mass-balance models and projections. *Journal of Glaciology*. https://doi.org/10.1017/jog.2019.22

Huybrechts, P., de Nooze, P., & Decleir, H. (1989). Numerical modelling of glacier d'Argentière and its historic front variations. In J. Oerlemans (Ed.), *Glacier Fluctuations and Climatic Change* (pp. 373–389). https://doi.org/10.1007/978-94-015-7823-3_24

Jouvet, G., & Huss, M. (2019). Future retreat of Great Aletschgletscher. *Journal of Glaciology*, *65*(253), 869–872. https://doi.org/10.1017/ jog.2019.52

Jouvet, G., Huss, M., Blatter, H., Picasso, M., & Rappaz, J. (2009). Numerical simulation of Rhonegletscher from 1874 to 2100. *Journal of Computational Physics*, *228*(17), 6426–6439. https://doi.org/10.1016/j.jcp.2009.05.033

Jouvet, G., Huss, M., Funk, M., & Blatter, H. (2011). Modelling the retreat of Grosser Aletschgletscher, Switzerland, in a changing climate. *Journal of Glaciology*, *57*(206), 1033–1045. https://doi.org/10.3189/002214311798843359

Le Meur, E., Gagliardini, O., Zwinger, T., & Ruokolainen, J. (2004). Glacier flow modelling: A comparison of the Shallow Ice Approximation and the full-Stokes solution. *Comptes Rendus Physique*, *5*(7), 709–722. https://doi.org/10.1016/j.crhy.2004.10.001

Le Meur, E., & Vincent, C. (2003). A two-dimensional shallow ice-flow model of Glacier de Saint-Sorlin, France. *Journal of Glaciology*, *49*(167), 527–538. https://doi.org/10.3189/172756503781830421

Letréguilly, A., & Reynaud, L. (1989). Past and forecast fluctuations of Glacier Blanc (French Alps). *Annals of Glaciology*, *13*, 159–163.

Marzeion, B., Hock, R., Anderson, B., Bliss, A., Champollion, N., Fujita, K., et al. (2020). Partitioning the Uncertainty of Ensemble Projections of Global Glacier Mass Change. *Earth's Future*, in press. https://doi.org/10.1029/2019EF001470

Schmeits, M. J., & Oerlemans, J. (1997). Simulation of the historical variations in length of Unterer Grindelwaldgletscher, Switzerland. *Journal of Glaciology*, *43*(143), 152–164.

Schmidt, L. S., Aoalgeirsdóttir, G., Pálsson, F., Langen, P. L., Guomundsson, S., & Björnsson, H. (2020). Dynamic simulations of Vatnajökull ice cap from 1980 to 2300. *Journal of Glaciology*, *66*(255), 97–112. https://doi.org/10.1017/jog.2019.90

Schneeberger, C., Albrecht, O., Blatter, H., Wild, M., & Hock, R. (2001). Modelling the response of glaciers to a doubling in atmospheric CO2: a case study of Storglaciären, northern Sweden. *Climate Dynamics*, *17*, 825–834. https://doi.org/10.1007/s003820000147

Stroeven, A., van de Wal, R., & Oerlemans, J. (1989). Historic front variations of the Rhone Glacier: simulation with an ice flow model. In J. Oerlemans (Ed.), *Glacier Fluctuations and Climate Change* (pp. 391–405). Kluwer Academic Publishers.

Vincent, C., Vallon, M., Reynaud, L., & Le Meur, E. (2000). Dynamic behaviour analysis of glacier de Saint Sorlin, France, from 40 years of observations, 1957-97. *Journal of Glaciology*, *46*(154), 499–506. https://doi.org/10.3189/172756500781833052

Zekollari, H., Fürst, J. J., & Huybrechts, P. (2014). Modelling the evolution of Vadret da Morteratsch, Switzerland, since the Little Ice Age and into the future. *Journal of Glaciology*, *60*(224), 1208–1220. https://doi.org/10.3189/2014JoG14J053

Zekollari, H., Huss, M., & Farinotti, D. (2020). On the imbalance and response time of glaciers in the European Alps. *Geophysical Research Letters*, *47*. https://doi.org/10.1029/2019GL085578

Ziemen, F. A., Hock, R., Aschwanden, A., Khroulev, C., Kienholz, C., Melkonian, A., & Zhang, J. (2016). Modeling the evolution of the Juneau Icefield using the Parallel Ice Sheet Model (PISM). *Journal of Glaciology*, *62*(231), 199–214. https://doi.org/10.1017/jog.2016.13

---

## Author Comment (AC1) · 21 Jul 2020

**Answer to the Anonymous Referee #1**

Dear Referee #1

We would like to first thanks you for your positive and constructive comments on our work.

**Main comments:**

• Section 3 on the ice flow modelling contains a lot of content, which is standard in the Elmer Ice workflow and widely used. I don't see the point of recalling this again (in particular the set of equations) in the body of the paper. I suggest to replace it by appropriate references, or to move it in appendix or supplementary material to keep the paper concise and focused. Only what is specific to the Mer de Glace case should be kept in the paper's body (e.g. the choice of A relative temperate ice, the implementation of given fluxes on the domain borders, ...) Details about the 3D mesh should also be moved to appendix. Last, modelling details normally belong to 'Methods', so I suggest to move a very concise version of Section 3 (Ice flow) to Section 4 (Methods), and possibly give further details in appendix.

Concerning the ice flow description (section 3), we agree that a long description is not necessary here. We merged the two sections "ice flow model" and "methods" in a single section 3 "Methods". To ease readability, the first subsection that described the ice flow model was shortened with only the free surface equation kept. A reference to Gagliardini et al. 2013 indicates where the reader can find more information. The "mesh" subsection was removed from the main text and displaced to the supplementary materials.

• While you mention similar existing Full Stokes-based models, or global glacier models of the Alps in introduction, you make no link at all in the discussion. Typically, questions like "How do your results compared with some recent global modelling by Zekollari and al. or other FS model of individual large-scale glacier such as Aletsch?" or "Is the Mer de Glace expected to resist better than other neighbouring glaciers?" could be addressed in your paper. Comparisons with recent other models, and other glaciers of the Alps would put your results in a broader context. In abstract and conclusion, you write "To our knowledge a comparison to data at this detail is unprecedented": This statement is vague and unsupported in the paper.

A comparison with results from other models was added at the end of the discussion.

• I feel the reading of the paper can be made more efficient if the English was improved by a professional. Line 10-12 p6 is an example of phrasing with repetitions that can be improved.

The article was already corrected by a native English speaker but we were vigilant to improve the style of the text in order to ease the reading. Your suggestions were very much appreciated.

In particular we have (i) reduced the repetitions in the text and (ii) verified the necessity of the citations and reduced their numbers.

**Specific comments**

• I do not fully understand your choice of mass balance approach for the forecast simulation. A logical choice would be to simply compute the PDD everywhere instead of doing so at the Tacul gate only, and extrapolating the MB via Eq. (7). PDD can computed everywhere assuming vertical lapse rate for temperature. By doing so, you would release your hypothesis of linearly changing mass balance with respect to z. Don't you lose any information by this assumption? You could possibly tune PDDs to fit observation at Tacul in the past.

We first want to say that above the Tacul gate, when we integrate the SMB over the accumulation area, we use the vertical temperature gradient given by the climatic scenarios (meteorological variables are given every 300m). Below the Tacul gate we choose another approach, as explained in the Methods section: we calculated the SMB at Tacul gate and used the same vertical gradient as for the hindcast, mainly for consistency. This vertical gradient is derived from in-situ observations (ablation stakes, see Fig. S1).

To check the validity of this gradient, we calculated the SMB gradient for all the SMB scenarios (SAFRAN and all GCM-RCM couples). The Fig. R1-1 present the vertical gradient for SAFRAN reanalysis and the run CLM_HadCEM_RCP45, which is representative of most of our simulations. The altitude of interest (1650 to 2250 m a.s.l.), in bold, shows an interannual variability of the gradient that we didn't take into account.

We didn't test the sensitivity to the interannual variability but we calculated for each year the difference of SMB between the Tacul and the front for SAFRAN. The difference of SMB is 9.7 m a$^{-1}$ ± 0.9 m a$^{-1}$. The standard deviation is low and we assume an averaged value would lead to similar results.

[Figure]

[Figure]

**Fig. R1-1**: *Vertical SMB gradient extracted from a) SAFRAN and b) a climatic scenario with our PDD method every 300 m. Meteorological variable are available every 300 m of altitude from 1500 to 3600 m a.s.l.: the gradients are calculated between each level and levels corresponding to the altitude of Mer deGlace are in thick lines. The example is given here for CLM_HadCEM_RCP45.*

Nevertheless, we explored the sensitivity to the gradient of SMB. For SAFRAN the averaged gradient is 0.007 a$^{-1}$ (i.e. 7 m a$^{-1}$ per km), lesser than the adopted value of 0.009 a$^{-1}$. For the climatic simulations, the value above 1800 m a.s.l. are similar the adopted value of 0.009 a$^{-1}$, below 1800 m a.s.l. the gradient is lower. We calculated the gradient between the Tacul gate and the front: from 2015 to 2050 the averaged gradient is 0.007 a$^{-1}$. The choice of these different gradients (between 0.007 a$^{-1}$ and 0.009 a$^{-1}$) leads to a maximal difference of SMB at the front of 1 m a$^{-1}$ where SMB is up to -12 m a$^{-1}$.

We performed three set of simulations, with the gradient shown in the article (db/dz=0.009 a$^{-1}$), with a lower gradient for the forecast (db/dz= 0.007 a$^{-1}$ after 2015) and with this lower gradient (db/dz= 0.007 a$^{-1}$) all along the simulations. In Fig. R1-2 we show the evolution of the velocity, thickness and front evolution for the three gradient scenarios. The differences are low, except for the evolution of the front where the SMB are the highest, especially when SMB are different since 1979.

Compared to the chosen scenario, with the lower SMB, Montenvers and Echelets gate are ice free 5 years later, in 2050 the tongue is 250 to 500 m longer.

[Figure]

**Fig. R1-2***: Altitude, surface velocity and front evolution for three scenarios with different SMB vertical gradient of 0.009 $a^{-1}$ (solid lines); 0.009 for hindcast and 0.007 $a^{-1}$ for forecast (dotted lines); 0.007 $a^{-1}$ (dashed line). Average RCPs scenarios are plotted with thick curves and the extremes scenarios with thin curves.*

Eq (7) Something must be wrong with this equation. Also, you have not defined zs and zs = zsTAC.
Eq (7) was effectively wrongly written. We forgot a sign '+', it is **b(t) = b$_{tac}$(t) + k$_b$\*(zs -zs$_{tac}$(t)).**
For information, this equation is numbered Eq (3) in the new version of the manuscript.
The free surface elevation zs was already defined after the free surface equation, zs$_{tac}$ is now defined in the sentence that follow Eq (7).

• l27-30 p8 'relationship between ice flux at the gate and surface mass balance upstream of the gate': Could you elaborate your method (and possible its limitations) based on the two following questions: i) can your result of 11 years be interpreted as the average time for the ice to travel from the surface to the gate? ii) This '11 year' result is probably highly dependent of the glacier regime (advancing, retreating, highly retreating ...), which is fine if you modelled the retreat over the next 3 decades, but if this is the case, can you elaborate? and thus equivalently justify your sentence l30: "It is furthermore assumed that this relationship will remain valid in the future."
The location of the Tacul gate is less than 1 kilometer downstream the ice fall where ice flows from the accumulation basin in less than a year (at speed up to 700 m $a^{-1}$). Then ice velocities are decreasing but remain high. Consequently, our result of 11 years can be interpreted as the average time for the ice to travel from the glacier du Géant to the Tacul gate. This makes us confident that this relationship flux between the flux at Tacul and integrated SMB at higher elevation will remain valid in the future.

• 116 p12 'the modeled glacier is too thick and velocity too high': this surprise me, I would expect that it is too thick because too slow following simple mass conservation principle. How do you explain that ice can be both too thick and too rapid?

Fig 5. shows that even the hindcast thin few years too late after the early growth (before 1986).
It starts to thin in 1989 at Trelaporte and around 1995 at Echelets and Montenvers gates. Then the glacier slows down and the rate of observed velocity decrease is well reproduced. However, both thicknesses and velocities are overestimated between 1990 and 2010. The trends are correct but the simulated glacier is not as reactive to the changing regime (from thickening to thinning) as expected. This explains our conclusion arguing that some transient processes, probably at the base, are missing.

• I missed where you describe the 'sensitivity experiments' of Fig. 9 and 10 in your text? I understand that the goal is test the influence of the artificial flux condition at Tacul, which is important as you only model part of the glacier by lack of data. However, this should be explained in the body of the paper, and not being left alone in Fig. 9 and 10.

At the end of the 'Methods' section we added a subsection 'Sensitivity experiments' (now 4.3, see p12) that present these additional simulations.

**Technical comments**

Abstract I suggest to start straight the abstract with the 'Mer de Glace' and remove general statements (i.e. the 3 first sentences) to make the abstract more focused.

We considered with attention your suggestion. Finally, we decided to keep the 3 first sentences. We made small changes changes that considered all the comments made by the three reviewers.

l 6-7 p1 'Mer de Glace (Mont Blanc area)' => 'Mer de Glace glacier, France'. Done. Now, in the rest of the text we refer to the glacier as "Mer de Glace".

Abstract As said earlier, 'To our knowledge a comparison to data at this detail is unprecedented.'
is (too?) vague statement and not supported.

As answered earlier, we add a paragraph in the discussion to support this statement. What we considered as 'unprecedented' in this study is a comparison of geometry and dynamics with yearly in situ observations.

Abstract It is hard to evaluate how much represent 2 to 5 km without having any prior idea of the glacier size, could you instead use (or add) a relative metric? i.e. 30% to 65% of the glacier length.

We added the total length of the glacier. We didn't give relative length retreat as it depends whether we use the total length (20 km) or the modeled one (7 km).

l 21 p1 Suggest changing 'fluctuations' into 'evolutions'. Done.

l 4 p2 'Process-based model were also developed to take into account simple dynamics (...)' gives the feeling that first physically-based ice flow models started in 2015, which is obviously not the case. Could you reference early SIA-based models applied to mountain glaciers to fix this?

A reference to Le Meur et al. 2003 was added.

l 8-14 p2 You should clearly write here that by 3D physical model, you mean Stokes without any shallow-like approximations that would reduce the dimension of the equations (unless I'm mistaken?).

We added a mention to "Stokes ice flow" solution to clarify the sentence.

l 10-14 p2 Can you reduce the number of citations? Do you need all of them? It is anyway not exhaustive.

We reduced the number of citations. To illustrate the "model describing the Stokes ice flow solution for the complex three-dimensional geometry of a whole glacier has become much more affordable" we kept Jouvet and Funk 2014, Réveillet et al. 2015, Gilbert et al. 2018, and for 'Elmer/Ice has already been used for a number of glacier applications' we kept Gagliardini et al. 2011, Réveillet et al. 2015, Gilbert et al. 2018.

19-21 p2 'In addition, running simulations ... over three decades.' This is crucial here to justify the use of 3D Full Stokes to capture the complexity of the Mer de Glace ice flow, however, this is not exactly what this sentence is doing. I suggest to reformulate it to emphasize that using such a state-of-the-art model is not just an additional opportunity, but necessary to model the complex ice flow. Otherwise the reader might wonder why a simple model cannot do the job.

We reformulated the sentence to justify the use of 3D Full Stokes:

*"In addition, running simulations on this glacier provides the opportunity to fulfill the need to capture with a Full Stokes ice flow model the local complexity of ice dynamics of a glacier that presents a large expansion before the 1980 followed by a rapid retreat over three decades."*

l 21 p2 'This dynamics make it necessary to take into consideration the delay in the glacier response to climatic forcing.' This justifies to take into account the ice dynamics, but not to use a sophisticated model. Just make sure the argument is used for the right reason.

Yes, this justifies to take into account the ice dynamics. Actually, we show in this paper that, even with a sophisticated model, discrepancies remain and we conclude that "the small differences between the model and the observations arise from assuming a constant basal friction parameter". It even lets some space to further inquiry the processes involved and we think the sentence is then justified.

l 31-32 As shown in Fig. 1, Mer de Glace glacier has a single tributary glacier, which is named Leschaux glacier. Text modified.

l5 p3 'maintained' => 'that are monitored' Done.

l6 p3 I expected to have the elevations given. monitored' Done.

l8 p3 ... they were not ... Text modified.

l11 p3 'paucity' =>) 'lack' Done

l13 p3 .. the model domain was restricted Text modified.

l14-15 p3 A transition causality word to connect the two last sentences is missing.
we added a transition causality word

l19-20 p3 Again can you reduce the number of citations?
We reduced the number of citations and kept Farinotti et al. 2017, Gilbert et al. 2020.

l29 p3 Rheological parameter with a constant value assuming temperate ice (A = 5.0159 e  24), do you mean 10  24 ? Could you give a reference for this value? Possibly consider change the unit into 158 MPa  3 a  1. This is twice the value of Cuffey and Paterson (2010).
Concerning the rheological parameter, A, a reference was added (Paterson 1994) and we modified the unit to $MPa^{-3} a^{-1}$.

l 8-9 p6 The two next sections describe in detail the respective boundary conditions for the two steps (hindcast and forecast) defined at the surface (mass balance) and at the Tacul and Leschaux gates (ice flux from the accumulation areas), respectively.
The original sentence was replaced by your proposition.

l10-12 p6 4 times 'surface mass balance' in 3 lines, can you improve the text? (or use an acronym)
We modified the text, rewriting a sentence and removing the unnecessary 'surface mass balance' repetitions.

l 8 p7 Define RCP and shortly comment of the meaning of the 3 pathways.
We added the term "Representative Concentration Pathway" and defined it in a new sentence.

*The 26 climate projections used here span the 3 Representative Concentration Pathway (RCP) (see Table 1 in Supplementary Material). Each RCP refers to a radiative forcings scenario considered by the IPCC and depending on the future volume of greenhouse gases emitted. They are labeled after the radiative forcing values by the year 2100 (RCP2.6, RCP4.5 and RCP8.5 corresponding to 2.6, 4.5 and 8.5 W $m^2$ respectively).*

l10 p7 'good performance', this statement calls for ad hoc references.
The 'good performance'of the degree day model is justified by a reference to Réveillet et al. 2017 who compared several SMB models with data including data from Mer de Glace.

l17 p7 'is underestimated by climate models': Do you mean in general or for the climate model you specifically use?
We replaced 'by climate models' with 'some reanalysis datasets' as it concerns the dataset we use. The sentence describing this dataset was rewritten to be more specific.

l21-22 p 7 For each scenario, the correction factor for precipitation is evaluated over the past period 1979-2015. 'historical' = 'past'

l 38 p10 'Fig.4.' => 'Fig. 4.'
The missing space was added.

l6 p8 'using the diagnostic formulation for SIA' => 'using the SIA'. Why qualifying the SIA formula as diagnostic?
Text modified.

9) and (10) Implementing Eq (10) straight without any correction sounds hazardous as the chance to get unrealistic velocities is high as the SIA model is local (sensitivity w.r.t rs) and not tuned. In contrast, adding a correction factor that multiply Eq. (10) and is tuned to ensure continuity in time between Eq. (9) and (10) would make sense.

Eq. (9) and (10) are now numbered Eq. (5) and (6). We did not implement any corrections in Eq. (10). The vertical distribution (as the horizontal one) do not change when moving from Eq. (9) to (10). The Tacul surface elevation and velocity is the same for the hindcast and the forecast in 2015 as the flux evolution at Tacul gate is relative to 2015 (see Fig. 2).

Eq. (10) As you know the thickness here, did you consider implementing the SIA with a "shape factor" that accounts for lateral resistance of the ice flow as function of the type of profile?
Considering that we knew the real geometry of the gate and an observed surface velocity field we decided to use these data instead of a shape factor for the hindcast. Fig. R1-3 shows our velocity distribution for 2003. For Eq. (10) we could have implemented the SIA with a "shape factor" but we, as we know the real geometry, preferred to keep the same velocity distribution for consistency for hindcast and forecast.

[Figure]

**Fig. R1-3**: *Velocity distribution at the Tacul gate of 2003.*

l14 p8 'his maximum central value of the velocity', do you mean 'maximum value along the transect'?
Yes indeed, the sentence was modified in:
"In all cases,  we assume that the form of the **vertical profile of the horizontal velocity normal to the flux gate**  is given by the Shallow Ice Approximation…"

l15 p8 'above transverse velocity profile': refer to Eq. (9).
We changed the reference for the good equation (old Eq. (9)., now Eq. (6)).

l16 p8 'Despite the differences in the methods' I'm not sure what are the 2 methods you referring: Eq. (9) vs Eq. (10)? Please, clarify.
As this point, we were comparing our method with the method that Berthier and Vincent (2012) used to infer the flux at Tacul.

l28 p8 Repeat the reference to Durand for the Safran reanalysis. Done.

l15 p11 It can be surprising to see so tiny differences between different RCPs. You may say that this was expected as RCPs differ substantially more after 2050 (if I'm not mistaken).
Indeed, the RCPs differ after 2050, a new sentence was added:
    "*Large differences between the pathway scenarios appear only after 2050 (not shown)*"

l 30 p11 "However, these strong differences in ice thickness and ice flux at the Tacul gate lead to much smaller absolute differences in thinning and ice flow velocity downstream of the gate; the lower the cross section, the smaller the response differences for the different scenarios." If you were normalizing the model discrepancies by the ice thickness or ice velocity at the gate, the differences would probably be more uniform, isn't it?

We plotted the normalized model/observation discrepancies by the ice thickness or ice velocity at the gate in the Fig. R1-4. The normalized thickness discrepancies are similar for the three gates. Discrepancies at Montenvers gate have a greater amplitude which may be due to the low thickness and largest sensitivity to SMB evolution. The same behavior seems to occur for the velocity discrepancies but observation at the Montenvers and Echelets are sparse.

[Figure]

**Fig. R1-4***: Thickness and velocity differences (model-observation ) normalized by the observations for the 3 lower gates.*

l1 p12 'assess' => 'estimate' or 'forecast' or 'predict'
'assess' => 'estimate'

l5 p13 'which is low relative to the mean ablation measured on the tongue of Mer de Glace' can you give a number? The ablation at Tacul gate is given in the Fig. 2a. The ablation measurements are provided in the supplementary materials. It corresponds to ~4 m a$^{-1}$ at Tacul and more than 10 m a$^{-1}$ at Montenvers.

l6 p13 'Other uncertainties arise from the linear extrapolation of ablation over the tongue (Eq. (7)) based on measurements in an area of clean ice', your assumption of linear MB w.r.t altitude might by the highest source of uncertainty? If yes, it would make sense to start with this.

We think there is two sources of uncertainties in our hindcast. One can result from the dynamics (most probably the constant sliding coefficient), and the other is attributed to our SMB model. Concerning the SMB we decided

to choose a simple approach as Réveillet et al. 2017 showed that a classical degree-day model was sufficient to simulate the long-term glacier-wide MB.

For the hindcast we had the choice to use a PDD using SAFRAN reanalysis (with meteorological variables available every 300 m of elevation) or the observations. We answered earlier the reason why we choose a linear MB w.r.t altitude.

The tongue of the Mer de Glace has an increasing debris cover, which is until now, restricted to the last kilometer downstream the Echelets gate. We assume that it partially protected the front during the two last decades. While it is difficult to know how the debris cover will evolve above Echelets gate in the future we have not taken this parameter into consideration in the siulations. However, we are conscious that it could lead us to overestimate front retreat.

l8 p13 Please reference to a Figure.
We added a reference to the (now) Fig. 5.

l10 p 13 'is also well known' => 'is also well constrained'. Corrected.

l11 p 13 not underestimated by the model... But Fig. 5 shows a slight but real systematic over-estimation of velocities, isn't?
As mentioned in the answer to Specific comments, Fig 5. shows that velocity increase during the glacier growth before 1986 is underestimated. Then the glacier slows and the rate of observed velocity decrease is well reproduced by the model at the three gates. These changes follow the thickness evolutions. Both thicknesses and velocities are overestimated between 1990 and 2010: thickness because the simulation seems to thin few years too late and velocity because the glacier is too thick.

l12 p 13 To what 'discrepancies' do you refer? In the sentence before, you say there is no major discrepancies in velocities.
The trends are correct, the model reproduce the growth and then the decay of the glacier but the simulation is not as reactive to the evolution at Tacul gate as expected by observations.

l1-4 p14 One can also argue an inaccurate sliding ratio to explain model errors? Have you considered testing different rate factors A to increase or decrease vertical deformation?
We didn't test different rate factors A. Indeed, the uncertainties on this factor may explain some model errors.

l13 p15 'will start to increase ...' => 'is expected to increase ...', check not to use future tense when you describe model results, but conditional or equivalent. Same for the sentence right after.
Tense were corrected in both sentence.

**Figures:**

Concerning the figures, all your suggestions were taken into account:

Fig 2. SMB not defined. How is it possible that different RCPs show different values in the past times?
SMB was defined in the caption for the level of Fig 2a.
In the past times the mean RCPs show different values as does not comprehend the same number of model (for instance there is 3 models for RCP 2.6, 11 models for RCP 4.5 and 13 for RCP 8.5.

Fig 4. Formulation (Eq. (10)),
Fig 4. observed surface velocity, ice thickness, and a velocity distribution given
Legend of Fig. 4 was corrected.

Fig. 6 "Same as Fig. 5, but for surface elevation." Please make a full sentence. For convenience, better to repeat the text to make each individual caption self-contained. Another possibility is to merge Fig. 5 and Fig. 6, with two panels.
The same for Figs. 9 and 10.
Fig. 5 and Fig. 6, as well as Figs. 9 and 10 were merged.

Fig. 8 colorbar is missing
Fig. 8: a colorbar was included.

Fig. 10 Color coding missing. Again, the Figure would be better if self-contained without having to look for the color coding in another Figure. Otherwise merge Fig. 9 and 10 with two panels.
Figs. 9 and 10 were merged.

Vincent Peyaud, on the behalf of the coauthors.

**Numerical modeling of the dynamics of Mer de Glace glacier, French Alps: comparison with past observations and forecasting of near future evolution.**

Vincent Peyaud[1], Coline Bouchayer[1,2], Olivier Gagliardini[1], Christian Vincent[1], Fabien Gillet-Chaulet[1], Delphine Six[1], and Olivier Laarman[1]

[1]Univ. Grenoble Alpes, CNRS, IRD, Grenoble INP, IGE, 38000 Grenoble, France
[2]Department of Geosciences, University of Oslo, 0316 Oslo, Norway

*Correspondence to:* Vincent Peyaud (vincent.peyaud@univ-grenoble-alpes.fr)

Correction in response to the reviews are in blue.

**Abstract.**

 Alpine glaciers are shrinking

[revised manuscript text omitted]

25    either on past or future evolutions. To validate the hindcast studies, the authors generaly compare length reconstruction and area or volume and area when DEM are available (Jouvet et al., 2011; Zekollari et al., 2014) Our studies allows a yearly comparaison of thickness and velocity changes at different locations of the glacier. The iconic status of the Mer de Glace and facilites of the Electricité De France for hydropower resources and Compagnie du Mont Blanc for tourist activities justifiy a specific study with a state-of-the art model. Touristic facilites will have to modified.

30    Vincent et al. (2014) used a parametrized model calibrated with past thickness changes to simulate the future fluctuations of the MdG. They found that the MdG will retreat by 1200m until 2040. Zekollari et al. (2019) used a SIA model to reconstruct the evolution of all glaciers located in the Alps. They also used EUROCORDEX ensemble scenarios. They found that between 2017 and 2050 Mer de Glace will retreat from 2 to 6 km which is close to our results.

Jouvet and Huss (2019) lead a forecast for Great Aletsch Glacier with a full stokes model and the EUROCORDEX ensemble,

35    the largest glaciers in European Alps, their results, also similar to Zekollari et al. (2019), predict a glacier retreat by around 5

km between 2017 and 2050.

This future evolution can also be related to the glacier response time which has been a subject of interest of several studies. The response time of glaciers are affected by several predictors such as the glacier size, the glacier SMB and glacier slope which is the main driver. Recently, Zekollari et al. (2020) used large scale glaciers modelling to investigate the response time in the European Alps. For Mer de Glace the relation proposed by the authors gives a response time of 60 years, which is typical of the range of European Alps glaciers ($50 \pm 28$ years). In our sensitivity experiment the future simulation performed with values flux at Tacul and SMB stationary 
[revised manuscript text omitted]

---

## Author Comment (AC2) · 21 Jul 2020

**Answer to the Anonymous Referee #2**

Dear Referee #2,

We thank you for your review. Your numerous comments in the original text were very appreciated and improve the readability of the text.
We do not explicitly answer to the editorial suggestion (i.e. corrections of the text) in this response but we included all them in the corrected version. We respond here to the main comments you have mentioned in the notes you included in the PDF.

**Main structure changes of the manuscript**

As the ice flow description belongs to the methods, the structure of the manuscript was slightly modified. We merged the two sections "ice flow model" and "methods" in a single section 3 "Methods". To ease readability, the first subsection that described the ice flow model was shortened with only the free surface equation kept. A reference to Gagliardini et al. 2013 indicates where the reader can find more information. The "mesh" subsection was removed from the main text and displaced to the supplementary materials.
We indicate also the position of our corrections in the PDF that contains our modifications.

**Response to main comments**

The significance of this study stems in part from the iconic status of the Mer de Glace, its historical significance and its prominent modern role as a tourist destination. With little extra effort, the authors could interpret the significance of their results in this context and add some interest and value to the paper that a broad readership would appreciate.
We agree that the iconic status of the Mer de Glace make this study significant and may interest readers beyond the scope. Parts of the article are rather technical but we tried to improve the discussion to interpret the significance of our results and address a broader readership. As said, the structure was simplified and the model description was lightened in the same purpose.

Though not a show-stopper for publication, it would be nice to see at least some comment on the sensitivity of the results to (1) the large precipitation correction factor and (2) the treatment of ice flux from the Leschaux gate.

The correction factor for precipitation is important (+70%) but this value is not surprising as models use to underestimate the amplification of precipitation at high altitude. We consider the uncertainties on precipitation are within the range covered by the 26 climatic scenarios.

To calibrate our PDD model, we looked at different values for the melting factors and for the correction factor for precipitations. Once chosen the melting factors (values from Réveillet et al. 2017) we determined the correction factor for precipitations to best fit SMB observations at Tacul gate (see Fig 2.) and on the accumulation area above 3000 m a.s.l. (not shown in our article). The value of 1.63 was found for the SAFRAN reanalysis. For the GCM-RCM couples the averaged value was 1.70. These values (1.63 and ~1.70) are similar as the climatic scenarios are corrected for the Alps by the ADAMONT method (Verfaillie 2017) using the SAFRAN reanalysis.
On the tongue, the correction factor has few, if none, influences has the ablation dominates the SMB. Averaged SMB is ~-5 m a$^{-1}$ at Tacul and ~-10 m a$^{-1}$ at Montenvers gate.

We use this factor to calculate the SMB on the accumulation basin (i.e. above the Tacul gate). This factor influences the integrated SMB we use to calculate the flux at the Tacul. Nevertheless, as future fluxes at Tacul gate are imposed relatively to their value of 2015: the absolute values are normalized and have no direct influence on the flux.
Thus, the influence of the correction factor for the flux at Tacul gate is limited to the rate of flux variation. For instance, in a scenario of monotonic decrease of the SMB, the lower the correction factor, the sooner integrated SMB will become null. As shown in Fig. 2b that presents the 11 years moving average integrated fluxes above

Tacul gate, our future scenarios spread the whole possible trajectory from the constant flux up to zeros as earlier as 2030. So, we can conclude that the choice of this parameter does not influence our results.

The treatment of Leschaux gate is very similar to the Tacul's one but as the flux through Leschaux gate is one hundredth smaller (estimated in 2003) than the one at Tacul we can assume that the sensitivity to this boundary condition is negligible after the 2000's. It was may be less obvious in the 80's and we take this into account as we have implemented a similar behavior at both gates, the velocity and thickness at Leschaux being driven with the relative evolution of the velocity and thickness at Tacul.
We decided to avoid making the text more cumbersome with the description of this boundary condition briefly presented in the 'Methods' (see l.7-15 p.9).

I have made numerous comments (the review effectively) in the attached pdf that I hope will assist the authors with revisions, including editorial suggestion intended to improve the readability of the text.
We really appreciated your editorial suggestions that improved the readability of the text.

**Line by line answer of the along-the-text comments**

Abstract: Too sweeping. If referring to only European Alps, write "Alpine". Not all glaciers worldwide are shrinking and retreating.
We modified the first sentence by *"Alpine glaciers are shrinking and rapidly lose mass in a warming climate".*

Abstract: "We found that the model accurately reconstructs the velocity and elevation variations of this glacier despite some discrepancies that remain unexplained." Can this be quantified or made more precise in some way?
We did not see which result would be the most significant and representative of the "accuracies" and/or "discrepancies" to be included in the abstract.

l16 p2: seems reasonable just to use the name of this glacier "Mer de Glace" rather than adding "glacier" at the end.
Excepted the first occurrence in the abstract, in the main text, we systematically refer to the glacier name as "Mer de Glace" without adding "glacier" at the end.

l16 p2: "as the glacier presents a large expansion before the eighties followed by a rapid retreat over three decades".
You suggested to modified the sentence ("presents a large expansion"). This is the rare suggestion we did not follow.

l6 p3: would be nice to give elevations here
(see l.10 p.3) We added the elevation of the four profiles.

L16 p3: The ice-flow model seems part of methods, especially since Mer de Glace-specific information appears in this section. & l1 p6: See earlier comment: ice-flow model seems it should be part of methods. SMB and input flux seem more like "Inputs" or "Boundary conditions" than a comprehensive description of study "Methods".
As answered earlier, we have modified the structure of the document. We merged the two sections "ice flow model" and "methods" in a single section 3 "Methods".

l16 p3: reference? Cuffey and Paterson? "e" is kind of junky notation for a paper. Suggest \times 10.
The reference for viscosity (Paterson 1994) was added, and the unit has been changed and $A=158$ MPa$^{-3}$a$^{-1}$.

p6: Eq (7) something seems wrong with this equation. Intercept is missing and balance gradient (db/dz) should be dimensionless, not [1/m].
(old Eq (7) now Eq (3)) was effectively wrongly written. We forgot a sign '+', it is $b(t) = b_{tac}(t) + k_b*(zs - zs_{tac}(t))$.

l10 p7: The lack of sophistication of the classical DD model seems out of step with the sophistication of the ice-flow model and the volume of data available. I'm not suggesting this be changed here, but some better justification (aside from "simplicity and relatively good performance) should be provided. Has an enhanced temp-index model been tried (eg Hock, 1999) and shown not to yield improved performance? Do the timescales involved warrant use of the simplest possible SMB model?
We added a reference on the works of Réveillet et al. 2017 on which we rely. Their "results support the use of a classical DD model for long-term simulations of glacier-wide MB results support the use of a classical DD model

for long-term simulations of glacier-wide MB". Three decades (until 2050) may be considered as shorter term, but the evolution meteorological variables will be unknown, in particular local wind, turbulent flux, etc. which increase uncertainties.

l15 p7: Is there a rain-to-snow threshold used?
The rain-to-snow threshold is based on temperature and is 0°C.

l32 p7: In all cases, for the vertical evolution over the artificial boundary, we assume that the form of the vertical profile of the normal velocity is given by the Shallow Ice Approximation
See l24p7 the sentence has been changed to *"In all cases, we assume that the form of the vertical profile of the horizontal velocity normal to the flux gate is given by the Shallow Ice Approximation."*

Figure4: In the legend, "SIA diagnostic formulation" is replaced by "SIA".

l8 p11: Please quantify at least some of these subjective assertions of "good agreement" and "very well reproduced".
"*despite this good overall agreement*": "*good*" was removed.
We think that the following sentences describe the behavior of the model and the differences with the velocity and thickness observations:
       "*For example, the peaks of calculated surface elevation and velocities are reached with a delay of about 3 years at Trélaporte. On the lower cross sections, Echelets and Montenvers, the surface elevation did not show a significant increase between 1979...*"

"*the general trend of snout retreat is very well reproduced*":
We think that the following sentence describes the behavior of the model and the differences with observations of the front:
       "*The simulated front is almost stable between 1979 and 1990 and starts 10 to retreat slowly 5 years before the rapid observed retreat in 1995. Over the period 1995-2015, the observed rapid retreat of the ice front is well reproduced with a retreat rate of 30 m $a^{-1}$ compared to 35 m $a^{-1}$ for the observations.*"

l22 p11: gate = cross sections = profiles, right? Choose one.
This is right. We replaced 'cross sections' and 'profile' by 'Gate' (see l.12 to 14 p.11).

Discussion: I have to say that I'm somewhat disappointed not to read anything about the social implications of the demise of MdG. One of the reasons this study is valuable is because of the iconic status, historical significance and tourist attraction of MdG. Can you say anything about, e.g., what the results might mean for the local community or how much more difficult it will be too access the MdG from Montenvers than it was in historical times, or whether this makes the ski tour from the AdMidi more/less safe, doable, etc?
We'll add a new paragraph in the discussion to compare our results with other studies (not finalized).
The retreat of the glacier with have an impact on the access to the tongue, and some hydropower or touristic facilities will have to be modified. We'll try to add a sentence on the local impact for the users of the Mer de Glace if possible.
Concerning the skitour, the increasing ablation will probably prevent use during late spring. It is difficult to predict the evolution of the ice fall without simulating the glacier du Géant and conclude on its influence on skitour feasibility.

Vincent Peyaud, on the behalf of the co-authors.

**Numerical modeling of the dynamics of Mer de Glace glacier, French Alps: comparison with past observations and forecasting of near future evolution.**

Vincent Peyaud[1], Coline Bouchayer[1,2], Olivier Gagliardini[1], Christian Vincent[1], Fabien Gillet-Chaulet[1], Delphine Six[1], and Olivier Laarman[1]

[1]Univ. Grenoble Alpes, CNRS, IRD, Grenoble INP, IGE, 38000 Grenoble, France
[2]Department of Geosciences, University of Oslo, 0316 Oslo, Norway

*Correspondence to:* Vincent Peyaud (vincent.peyaud@univ-grenoble-alpes.fr)

Corrections in response to the reviews are in blue.

**Abstract.**

 Alpine glaciers are shrinking

[revised manuscript text omitted]
 have been performed by other researchers. Most studies focused either on past or future evolutions. To validate the hindcast studies, the authors generaly compare length reconstruction and area or volume and area when DEM are available (Jouvet et al., 2011; Zekollari et al., 2014) Our studies allows a yearly comparaison of thickness and velocity changes at different locations of the glacier. The iconic status of the Mer de Glace and facilites of the Electricité De France for hydropower resources and Compagnie du Mont Blanc for tourist activities justifiy a specific study with a state-of-the art model. Touristic facilites will have to modified.

Vincent et al. (2014) used a parametrized model calibrated with past thickness changes to simulate the future fluctuations of the MdG. They found that the MdG will retreat by 1200m until 2040. Zekollari et al. (2019) used a SIA model to reconstruct the evolution of all glaciers located in the Alps. They also used EUROCORDEX ensemble scenarios. They found that between 2017 and 2050 Mer de Glace will retreat from 2 to 6 km which is close to our results.

Jouvet and Huss (2019) lead a forecast for Great Aletsch Glacier with a full stokes model and the EUROCORDEX ensemble, the largest glaciers in European Alps, their results, also similar to Zekollari et al. (2019), predict a glacier retreat by around 5

km between 2017 and 2050.

This future evolution can also be related to the glacier response time which has been a subject of interest of several studies. The response time of glaciers are affected by several predictors such as the glacier size, the glacier SMB and glacier slope which is the main driver. Recently, Zekollari et al. (2020) used large scale glaciers modelling to investigate the response time in the European Alps. For Mer de Glace the relation proposed by the authors gives a response time of 60 years, which is typical of the range of European Alps glaciers ($50 \pm 28$ years). In our sensitivity experiment the future simulation performed with values flux at Tacul and SMB stationary 
[revised manuscript text omitted]

---

## Author Comment (AC3) · 21 Jul 2020

**Answer to the Anonymous Referee #3**

Dear Referee #3

We would like to first thank you for your positive and constructive comments on our work.

**Main structure Changes of the manuscript**

As the ice flow description belongs to the methods, the structure of the manuscript was slightly modified. We merged the two sections "ice flow model" and "methods" in a single section 3 "Methods". To ease readability, the first subsection that described the ice flow model was shortened with only the free surface equation kept. A reference to Gagliardini et al. 2013 indicates where the reader can find more information. The "mesh" subsection was removed from the main text and displaced to the supplementary materials.
We indicate also the position of our corrections in the PDF that contains our modifications.

**General comments:**

It is a pity to see that only the lower part of the Mer de Glace was modelled (as explained in section 3.2 and 4.2). While going through the manuscript, I was constantly thinking: why is this the case? The explanation, in which this is linked to the uncertainty in the bedrock, appears only towards the end of the manuscript (p.15, l.20-21). If you were to consider the upper part of the glacier also, you would indeed introduce additional uncertainty in your simulations; but you now also do so by imposing several conditions on the fluxes through the upper gates (Tacul and Leschaux): e.g. linking the flux at the gates with upstream integrated surface mass balance based on observations and imposing this for the future (while in reality the glacier response time will play a big role here). It would really be nice to see how much this influences your results by having some additional sensitivity tests in which you modify the imposed model settings. Even better would be to have some simulations in which you model the entire glacier (i.e. after inverting the ice thickness in the accumulation area) and see how they compare to your results. Would this be feasible? Such a test would require some additional work, but I honestly think that this would add a lot to your story and would also increase the impact of your story (as it could be used as a kind of reference for future studies that impose fluxes at gates and only model a part of the glacier – a method that may definitely gain in popularity for certain applications!)

We agree it would be more satisfactory and impressive to have modeled the whole Mer de Glace up to its accumulation basin. Unfortunately, measurements of the bed are very parse upstream the Tacul Gate, and inexistent on the Lechaux Glacier. We choose to concentrate on the lower part where a large dataset exists.
By the way, strictly speaking, Mer de glace is restricted to the name of the tongue of the glaciers downstream the merging of Tacul and Leschaux glaciers. To alert the reader, in the introduction (see l.34 p.2) we added a mention to "the tongue" of the Mer de Glace:

> *"to reconstruct these past multi-decadal fluctuations **of the tongue of the Mer de Glace**".*

The Tacul boundary condition is well constrained by observation during the hindcast period. In the case of using a reconstructed dataset, uncertainties are also very large. For example, the reconstruction of Huss and Farinotti 2012 of the Argentière Glacier underestimate thickness locally by a factor two (Rabatel et al. 2018). Only a careful calibration of upstream surface mass balance and dynamics (viscosity/sliding) may constrain correctly the flux at Tacul. In all cases we have to inverse the sliding, using the observed velocity of 2003. Thus, we would have a couple thickness/sliding calibrated with one observation at Tacul gate. This calibration would be kept constant for the forecast simulations. The discussion of the relative uncertainties of the two methods could be a part of another study. We add a mention to this point in the perspective.

The level of detail of your analyses is very sophisticated and you consider several elements in your evaluation and for the projections that many other studies do not include.
  o However, it would also make sense to have insights in the more widely considered glacier

characteristics, such as glacier volume and area:

§ For past: with this you can directly assess the performance of your model to reproduce e.g.:

• Past volume changes (would in fact be a kind of test for your surface mass balance model in this case), which can be derived from DEM differencing
(and which I imagine is maybe already directly available for this glacier?).

• Past area changes.

[Figure]

Fig R3-1: Area and volume evolution for the hindcast simulation.

We plotted the area and volume evolution for the hindcast simulation (See Fig R3-1). Comparisons with the few MNTs available show a good agreement. The Fig. 5 showed that we overestimate the thickness. The new figure shows that the volume is also slightly overestimated (in 2003). The volume decreases between 1979 and 2003 (14% of the initial volume) is underestimated by 30%, extent reduction is also underestimated (10% of the initial area) is underestimated by 25%.
We will add this figure in the Supplementary Material.

§ For future: allows you to compare more easily to other studies in which the evolution of Mer de Glace is also modelled and comparison with other glaciers in the European Alps (e.g. is Mer de Glace more/less retreating than other glaciers...; list of studies is provided further on) In general, you refer to your study as an elaborate evaluation, which it definitely is, and which I think is very impressive. However, some of the agreement may also results from several choices you made, which are not always explained (see comment below on this). It is therefore difficult to disentangle which part of the agreement results from a kind of calibration ('tuning') and is therefore not a real kind of validation/evaluation (as you want the calibration and the evaluation data to be – ideally entirely – independent).
We added a comparison with results from other models in the discussion. We answer below to the point you have proposed to disentangle tuning from evaluation. We "acted" on the three boundary conditions (Flux at Tacul gate, sliding and SMB). No other choices influence the dynamics of the glacier at the four validation datasets (the three gates and the front).

• There is a sort of discrepancy between the complexity of the model used for the ice dynamics and the relatively rough approach for surface mass balance and imposed boundary conditions at the gates. Given that the glacier is so well studied, why did the authors not consider more complex approaches for this (e.g. thinking of e.g. debris cover; constant mass balance gradient)? A few additional sentences and motivation would be nice.
Yes, the use of an ice flow model with high complexity contrasts with the simple degree day approach for the surface mass balance (SMB) and the use of a constant and homogenous vertical gradient for SMB.
L.9 p.7 (now see l.9 p.7) we added a reference on the works of Réveillet et al. 2017 on which we rely. Their "results support the use of a classical DD model for long-term simulations of glacier-wide SMB". Three decades (until 2050) may be considered as shorter term, but the evolution of meteorological variables will be unknown, in

particular local wind, turbulent flux, etc. which increase uncertainties and render very complicated the use of a more sophisticated SMB model.

The Argentière glacier, situated also in the Mont Blanc moutain range is well more monitored, with an AWS on the side: it could be a good choice for future studies with more complex SMB model. But Mer de Glace cannot allow such validation of the SMB model. This is why we decided not to add debris cover influence in this study as it would be another "tuning" to explained the behaviour of the Montenvers evolution in the recent years (>2000). Below we add a sensitivity experiment on the SMB vertical gradient and also discuss the debris cover.

• Some assumptions are made, and it is not always clear how these affect your results. Would be good to have some additional insights in the sensitivity of your findings to your various assumptions. This includes assumptions related to:
    o Constant mass balance gradient
    o Imposed ratio between sliding and surface velocities at the Tacul gate
    o Assumption that relationship between ice flux at Tacul gate and integrated surface mass balance for upstream area remains the same
    o Minimal thickness and velocities at the gates
    o Linear decrease ice thickness at Leschaux gate over time,
For a full list and more details, refer to the specific comments below.
We answered your comment on these various assumptions in our responses below.

• Most of the figures could be improved relatively easily to enhance their readability: see suggestions below.
We modified the figures and describe the specific changes below.

**Specific comments and suggestions**

**Abstract:**

• p.1, l.2: 'All alpine glaciers are shrinking and retreating at an accelerating rate...': technically this is not entirely true. It is the case for most glaciers, but there are some exceptions (e.g. glaciers that are almost gone or those that disappeared; i.e. where the retreat does not accelerate). Suggest changing this to: 'Alpine glaciers are shrinking and rapidly lose mass in a warming climate'
We replaced the first sentence of the abstract by your proposition.

• p.1, l.8-9: 'To our knowledge a comparison to data at this detail is unprecedented': indeed, a very detailed comparison to data is present, which is very nice. But not sure you can claim that it is unprecedented, as comparisons to other studies are not straightforward (in some studies other types of data have been considered). Probably best to remove from abstract and mention in this in the main text, where there is room for more nuance. Check studies on individual glaciers with elaborate evaluation and/or calibration with ground-truth data (e.g. Adalgeirsdóttir et al., 2011; Zekollari et al.,2014; Hannesdóttir et al., 2015).
We wrote that a "comparison to data at this detail is unprecedented" because this is "to our knowledge" the first comparison of a model with geometry and dynamics from yearly in-situ measurements. Most studies compare length variation, or volume variation when MNT are available, but rarely long-term local observations. We explained this in a new paragraph (not finalized) at the end of the 'Discussion'.

• p.1, l.9-10: You mention the velocities and the elevation changes for the model evaluation. What about the mass balances and the length variation, which you mention a few sentences before (in l. 6): how do these perform? This becomes clear in the text, but for consistency would be good if you could already mention them here.
We added the fact that the model reproduces well the length variation.
By mass balance do you mean volume evolution? The mass balance is also well reproduced but we do not present it in our figures, so we do not mention it at this point.
The new sentence is:
    "*We found that the model accurately reconstructs the velocity, elevation and length variations of this glacier…*"

**1 Introduction:**

• p.1, l.19: 'sea-level rise': could be worth referring to the new GlacierMIP studies, in which the future sea level contribution from glaciers are obtained from a community-wide intercomparison effort (Hock et al., 2019; Marzeion et al., 2020)
References to the new GlacierMIP studies (Hock et al., 2019; Marzeion et al., 2020) have been added.

• p.1, l.21-22: 'first studies': you are not very specific here. Given that you model a single glacier and have not mentioned the 'large-scale' glacier modelling aspect yet, one would assume that these are the first studies for the evolution of individual glaciers in the European Alps. I suggest being more specific here (mentioning the regional aspect) and/or to refer to pioneering studies in which ice dynamics are included for individual glaciers (e.g. Huybrechts et al., 1989; Letréguilly & Reynaud, 1989; Stroeven et al., 1989; Greuell, 1992). Would somehow be strange to spend your introduction focusing on largescale glacier modelling, while your work in fact focuses on very detailed glacier modelling.
We introduced the first studies of an individual glaciers to start the sentence that introduce the different possible methods (last studies cited are large scale). We add in the list for each method the first study you mentioned but we kept also the last one. Here are the new sentences:
> *"The first studies of an individual glaciers (e.g. Huybrechts et al., 1989; Letréguilly and Reynaud, 1989; Stroeven; Greuell, 1992) were restrained to flowline model related to the local driving stress while studies on regional scale (since Hae- berli and Hölzle, 1995) focused on an empirical approach in which ice dynamics were not taken into account explicitly and glacier evolution was based on parameterization calibrated either on equilibrium-line altitude (ELA) model (e.g. Zemp et al., 2006), extrapolation of observed geometry changes (e.g. Huss et al., 2008; Huss, 2012; Huss and Hock, 2018) or volume and length–area scaling (e.g. Marzeion et al., 2012; Radic´ et al., 2014). Process-based model were also developed to take into account simple dynamics (e.g. Le Meur and Vincent, 2003; Clarke et al., 2015; Zekollari et al., 2019; Maussion et al., 2019)."*

• p.2, l.6: better also update with the new numbers from the second GlacierMIP effort (Marzeion et al., 2020)
The reference has been updated.

• p.2, l.10-12: list of references for 'model describing the complex three-dimensional geometry of a whole glacier' is a bit odd:
> o Some studies do not take into account the glacier evolution over time
> o Others are in fact based on the SIA, which makes them rather 2D (as described in the title of Le Meur & Vincent, 2003) and more in line with what you describe earlier (p.2, l.4-5) as
'Process-based model ... to take into account simple dynamics' (Clarke et al., 2015)
> o The ITMIX experiment, which focuses on ice thickness reconstruction (Farinotti et al., 2017), is also a bit odd to mention here
> o Why not simply focus on what you also do here: 3-D time-evolving simulation of a single glacier? (e.g. Schneeberger et al., 2001; Le Meur et al., 2004; Jouvet et al., 2009, 2011; Zekollari et al., 2014; Ziemen et al., 2016; Jouvet & Huss, 2019; Gilbert et al., 2020; Schmidt et al., 2020).
Would also be interesting for the discussion to compare your modelled future evolution of Mer de Glace with the modelled evolution of other glaciers in the European Alps (see also general comment on this).
We modified the sentence to focus on Full Stokes model and keep few of the most recent references:
> *Indeed, with the improvement of computational resources performance, running a model describing the* *Stokes ice flow solution* *for the complex three-dimensional geometry of a whole glacier has become much more affordable (e.g. Jouvet and Funk, 2019; Réveillet et al., 2015; Gilbert et al., 2020).*

• p.2, l.21: 'This dynamics' à 'These dynamics'
That sentence was removed and the previous one was rewritten (see l.22-24 p.2).

**3 Ice flow model:**

• p.3, l.29: Value for the rheological parameter for ice: how was this value chosen? Quite often this is used as a calibration parameter as it has a large influence on the ice thickness (/glacier volume). By just taking a value from literature: difficult to assume that this will work well immediately for your glacier of interest. See studies in which this was analysed / where this rheological parameter was tuned (e.g. Schmeits & Oerlemans, 1997; Albrecht et al., 2000; Vincent et al., 2000; Giesen & Oerlemans, 2010; Adalgeirsdóttir et al., 2011). From my understanding, in your study the calibration occurs through the basal sliding, in which you try to match observed velocities: but what is effect of this approach on modelled ice thickness evolution? i.e. How are you sure that the modelled evolution

is related to physical forcing and not to some kind of model drift? Would be good if you could explain this a bit in the manuscript.

The reference for viscosity (Paterson 1994) was added, and the unit has been changed and A=158 MPa$^{-3}$a$^{-1}$.

We did not explore other value of viscosity. In that case we will add to inverse a new sliding. This would change the vertical deformation. This is a limitation of our study. We will discuss this limitation in the discussion part of the manuscript.

We have verified that there was no special drift of the model (due to discrepancy sliding2003/topography1979) at the beginning of the simulation.

• p.4, Figure 1: would have been nice to have surface elevation information in this figure (vs. visual imagery). Through this, would be easier to orient for someone who's not very familiar with the glacier.

We did not add altitudes in this figure but we wrote the elevation of the 4 profiles in the main text.

• p.5, l.16: model domain does not cover the entire glacier. Why? (I saw later that this is explained towards the end of the manuscript) Should really clarify this choice. Pity to not have the entire glacier in / or additional experiments in which this is the case to compare to,

This is already described at the end of the 'study site' but maybe not enough visible: this sentence is now a new paragraph (L.23-26 p.3, with text unchanged).

• p.5, l.28-29: 'Bedrock elevation...interpolation (Fan et al., 2005) of all available observations': does this mean that the bedrock is simply obtained from a kind of kriging? Is it not justified to rely on a more sophisticated approach, especially given the fact that you then use a very complex 3-D model to solve for dynamics and temporal evolution? Would also be good to have an idea where the ice thickness (/bedrock elevation) was measured (unpublished data is mentioned later; but maybe you can add the profiles in a figure somewhere?).

Bedrock elevation is obtained by kriging but the dataset is dense. The mesh subsection was moved to supplementary and we will give the location of the different radar profiles used to build this DEM.

**4 Methods:**

• Name of the section ('Methods') is maybe not ideal, as in fact the previous section ('Ice flow model') is also really part of the methods. As you mainly describe the boundary conditions here (at the cross sections), you could consider renaming this section 'Boundary conditions' or something alike?

As said earlier the 'Ice flow model' was merged with the 'Methods' section.

• p.6, Figure 2:
      o Quite difficult to decipher this figure: the grey line, which represents the 'average' is barely visible in the right panel.
      o Not ideal to combine green and red colours for lines in a single figure, given that a considerable amount of people cannot see the difference between these two colours (see e.g. https://en.wikipedia.org/wiki/Color_blindness#Red%E2%80%93green_color_blindness).
      o One needs to look up in the caption what the average stands for, maybe specify that this is the average of the RCPs? Same of Safran: maybe good to specify this, as not clear what this is at this point in the manuscript (i.e. Reanalysis 'SAFRAN')
      o Do not entirely get why you show RCP's for the past and how this should be interpreted. Makes sense that these are off if they have not been forced with reanalyses product (e.g. ERA5). With this, expect them to be much closer to SAFRAN reanalyses product also. Also, not entirely clear if what you show here is the SAFRAN original SMB, or the one that is corrected by scaling the precipitation with ca. +60-70% (as you describe towards the end of section 4.1.). I expect the latter, given the good agreement in SMB. If so, and if I understand it correctly, would it also make sense to have the 'modified' SMB (with precipitation correction) from the RCPs?

This figure was improved. We modified the RCP colours. We changed the green color for SAFRAN to orange to avoid mixing green and red. The gray line was slightly enlighted and is a bit more visible.

Indeed, the SAFRAN and the forecast scenarios values are the one that are corrected by scaling the precipitation with ca. +60-70%. As (i) the legend is long, (ii) the precipitation correction is one parameter of the SMB model and (iii) that model is described latter in the text we decided to keep the same text. We fear it would be more confusing to the reader.

• p.6, l.5-7: 'For the forecast simulations from 2015 till 2050, results from climate simulations are used to evaluate the flux on the different boundary conditions of the glacier domain': I get the meaning of this sentence, but it is a

bit strange / misleading to use 'evaluate' here, as this is what you use to describe the evaluation of the hindcast also. Maybe change to: '...are used to simulate the future flux evolution at the boundary of the glacier domain'
Changed to: '...are used to simulate the future flux evolution at the boundary of the glacier domain'

• p.6, l.6: 'till' à 'until'
Done

• p.7, l.4-5: 'Despite this strong variability from year to year (Fig. S1 in the Supplementary Material and Rabatel et al., 2005), a constant mass balance gradient of kb = 0.009 m–1 is adopted for hindcast and forecast simulations'. How does this affect your results / how would your result look like if you take into account the interannual variability and also not rely on a constant gradient?

To check the validity of this gradient, we calculated the SMB gradient for all the SMB scenarios (SAFRAN and all GCM-RCM couples). The Fig. R3-2 present the vertical gradient for SAFRAN and the run CLM_HadCEM_RCP45, which is representative of most simulations. The altitude of interest (1650 to 2250 m a.s.l.), in bold, shows an interannual variability of the gradient that we did not take into account.

We did not test the sensitivity to the interannual variability but we calculated for each year the difference of SMB between the Tacul and the front for SAFRAN. The difference of SMB is 9.7 m a$^{-1}$ ± 0.9 m a$^{-1}$. The standard deviation is low and we assume that an averaged value would lead to similar results.

[Figure]

**Fig. R3-2**: *Vertical SMB gradient extracted from a) SAFRAN and b) a climatic scenario with our PDD method every 300 m. Meteorological variable are available every 300 m of altitude from 1500 to 3600 m a.s.l.: the gradients are calculated between each level and level at the altitude of Mer deGlace are in thick lines. Example given for CLM_HadCEM_RCP45.*

Nevertheless, we explored the sensitivity to the gradient of SMB. For SAFRAN the averaged gradient is 0.007 a$^{-1}$, lesser than the adopted value of 0.009 a$^{-1}$. For the climatic simulations, the value above 1800 m a.s.l. are similar to the adopted value of 0.009 a$^{-1}$, below 1800 m a.s.l. the gradient is lower. We calculated the gradient between the Tacul gate and the front: from 2015 to 2050 the averaged gradient is 0.007 a$^{-1}$, also lesser than the adopted value of 0.009 a$^{-1}$. The choice of these differents gradient (between 0.007 a$^{-1}$ and 0.009 a$^{-1}$) leads to a maximal difference of SMB at the front of 1 m a$^{-1}$ where SMB is up to -12 m a$^{-1}$.

We performed three sets of simulations, with the gradient shown in the article (db/dz=0.009 a$^{-1}$), with a lower gradient for the forecast (db/dz= 0.007 a$^{-1}$ after 2015) and with this lower gradient (db/dz= 0.007 a$^{-1}$) all along the simulations. In Fig. R3-3 we show the evolution of the velocity, thickness and front evolution for the three gradient scenarios. The differences are low, except for the evolution of the front where the SMB are the highest, especially when SMB are different since 1979.
Compared to the chosen scenario, with the lower SMB Montenvers and Echelets gate are ice free 5 years later, in 2050 the tongue is 250 to 500 m longer.

[Figure]

**Fig. R3-3**: *Altitude, surface velocity and front evolution for three scenarios with different SMB vertical gradient of 0.009 a⁻¹ (solid lines); 0.009 for hindcast and 0.007⁻¹ for forecast (dotted lines); 0.007 a⁻¹ (dashed lines). Average RCPs scenarios are plotted with thick curves, the extremes scenarios with thin curves.*

• p.7, l.6-9: why did you consider these 26 future climate projections from the EURO-CORDEX ensemble, given that there's many more (>50) available? Any criterion used to choose only those? Moreover, are these simulations from the RCM at 0.11° resolution or at 0.44° resolution (or a mix?). Would be good if you could be a bit more specific on this.

The 26 future climate projections from the EURO-CORDEX ensemble belongs to the ensemble of regional climate projection that was adjusted with the ADAMONT method (Verfaillie et al., 2017). The RCM chosen are at 0.11° resolution (≈ 12.5 km).

The adjustment is described in Verfaillie et al. (2018): the authors assume that the 13 GCM–RCM pairs reasonably sample the overall uncertainty resulting from the 3RCPs (2.6, 4.5 and 8.5), even though not all EURO-CORDEX GCM– RCM combinations are available. The EURO-CORDEX raw surface fields were adjusted using the ADAMONT method, which is a quantile mapping and disaggregation method taking into account weather regimes to provide multi-variable hourly adjusted climate projections.

In the section "Surface mass balance" of the "Methods" we rewrote the sentence that describes the source of these climatic scenarios:

>*"For future simulations, the surface mass balance at Tacul gate in Eq. (3) is inferred from a series of 26 downscaled and adjusted regional climate projection of the EURO-CORDEX program (Jacob et al., 2014). The adjustment was performed using the ADAMONT method (Verfaillie et al., 2017) using the SAFRAN reanalysis (Durand et al., 2009) as an observation reference, as described in Verfaillie et al. (2018). The 26 climate projections used here span the 3 IPCC scenarios Representative Concentration Pathway (RCP) RCP2.6, RCP45 and RCP8.5."*

• p.7, l.34 – p.8, l.1: 'we further assume a constant and uniform ratio between sliding and surface velocities of 1/3 at both gates': what is this assumption based upon? Given the lack of direct observations of basal velocities, the uncertainty on this statement is quite large. How does this influence your results?
The factor was inferred to fit Berthier and Vincent 2012 estimation of the flux. The methods allow to have a realistic and "continuous" velocity field through the boundary condition. Downstream (on the next grid cells) the 3D velocity field adjust with the sliding coefficient. We performed some tests but what is important for the simulation is the total flux through the gate.

The sensitivity to this factor is low: with our velocity distribution, between a ratio of 0.2 and 0.5 the flux at the gate differs by only 0.2%.

• p. 8, l. 28-30: assumption about relationship flux at Tacul glacier and integrated surface mass balance higher area. You assume that this remains constant in the future: how much does this influence your results?
We have assumed that the relationship flux at Tacul and integrated SMB at higher elevation does not change. Indeed, the location of the Tacul gate is just downstream the ice fall where ice flow from the accumulation basin in less than one year (at speed up to 700 m a$^{-1}$). We expect a very rapid response of Tacul flux to flux that come from accumulation basin.

The sensitivity experiments test the influence of the artificial flux condition at Tacul gate and show that in the next decades the tongue is more sensible to the local surface mass balance.

• p.9, l.7: imposed values for minimal ice thickness and surface velocities. Why are these values chosen? And what is influence on your simulations?
These values are arbitrary. They correspond to a flux at Tacul of $2.10^5$ m$^3$ a$^{-1}$ (to be compared with the flux in 2015: $1.10^7$ m$^3$ a$^{-1}$). The remaining minimal thickness at Tacul gate may stop the front retreat 300 m downstream the gate (6 grid points). However, as explained in the main text (see l.29-34 p.8) the minimal thickness (corresponding to the minimal flux) is reached by no scenarios by 2050. In few run a minimal value for the velocity is reached but we checked that this does not influence the front retreat. In our sensitivity experiments, we show that for the mean reference scenario, the integrated SMB (mass loss) is well above the flux though the Tacul gate (mass gain) in 2050 (see now Fig. 9). The former process dominates the evolution of the tongue. As mentioned in the text, this is why our forecast stop in 2050.

• p.9, l.14-15: linear decrease in ice thickness at Leschaux gate: again, sounds rather arbitrary. What is effect of this on your simulations? Can imagine that this could have quite an impact on projected future changes.
The effect of ice flux from Lechaux is negligible after 2015. As mentioned in the main text, we have estimated this flux with the 2003 observed surface velocity field. At this date the flux at Leschaux gate was two orders of magnitude lower than the one estimated at Tacul gate.

The influence of Leschaux tributary was one of our issue when we initiated this work but it occured that this gate had no significance on the result after the 2000's. It was may be less obvious in the 80's and we take this into account as we have implemented a similar behavior at both gates, the velocity and thickness at Leschaux being driven with the relative evolution of the velocity and thickness at Tacul.

• p.9, Figure 3 + p.10, Figure 4:
o Like for figure 2: would make sense to have information on what the different lines represent in the figure itself instead of in the caption. There is enough space, and would make the interpretation much easier and more intuitive (and an advantage if you plan on using this in a presentation later!)
Figures 3 and 4 were modified

**Results:**

•Although you do not really calibrate to this, it is not fully clear how the several choices you have made affected your simulations and can therefore be considered as an independent evaluation.
For the hindcast simulation, the parameters we calibrated are (i) the 2D sliding coefficient of the sliding law with the 2003 observed velocity field, (ii) the flux at the Tacul gate inferred from thickness and velocity in-situ observations and (iii) the surface mass balance given by ablation stakes measurements at Tacul gate and at different places on the tongue for the vertical gradient.

Surface mass balance and sliding calibrations include observations at the profiles where the model is evaluated but do not affect the transient ice flow response modeled by Elmer/Ice.

• p.11, l.1-2: 'In general, the lower the profile, the larger the delay between the start of decrease of the simulated compared to the observed surface elevation': this does not really come as a surprise, as you impose the fluxes at

the Tacul gates to fit observations. As such the uncertainty in your results 'spreads' as you go away from these points (towards lower glacier elevations).
Indeed, this sentence is just a description of the result.

• p.12, Forecast simulation: part of the similarity in future evolution is driven by the fact that the SMB is quite similar, as the difference in the forcing (temperature and precipitation) increases with time and in fact only becomes really notable during the second part of the 21st century. However, a part of this is also simply related to the glacier response time, which is in the order of decades for this glacier.
Would probably be worth placing this in context a bit (potentially in the discussion session) and making the link to response time studies on Alpine glaciers (e.g. Zekollari et al., 2020).
As the SMB are similar for all RCP until 2050 we clarify this point with a new sentence (l.14 p.11):
> "*Large differences between the pathway scenarios appears only after 2050 (not shown).*"

A new paragraph (not finalized) has been added in the discussion to compare our results with previous studies. News sentences can be included to introduce the response time:
> "*This future evolution can also be related to the glacier response time which has been a subject of interest of several studies. The response time of glaciers are affected by several predictors such as the glacier size, the glacier SMB and glacier slope which is the main driver. Recently, Zekollari et al., 2020 used large scale glaciers modelling to investigate the response time in the European Alps. Using the relation proposed by the authors Mer de Glace should react to climate change with a response time of 60 years, which is typical of the range of European Alps glaciers (50 +/- 28 years). In our sensitivity experiments, the future simulation performed with values flux at Tacul and SMB stationary after 2015 shows that the Mer de Glace will reach a steady state in ~70 years after a front retreat of 4.1 km.*"

• Fig. 5-7: would again be good if could read the figure without having to refer to the legend (i.e. add information about the RCPs in figure directly). For RCP colours, it would be nice if you could use more conventional colours for the different RCPs (e.g. those used in the IPCC reports). Finally, mixing green and red in the same figure is still not a very good idea.
We modified the RCP colours and we changed the green color to avoid mixing green and red.

**Discussion & conclusion:**
• Nicely elaborated and very interesting in general!
Thank you, we appreciate that you find this discussion interesting.

• p.13, l.6-8: role of debris is mentioned. Given the complexity of your ice flow model and the level of detail of your analysis, would it not make sense to incorporate debris cover in your approach (and eventual evolution over time; see e.g. Jouvet et al., 2011)? Or maybe do some tests in which this is incorporated in a parameterized way to analyse whether this decreases the discrepancy between observations and modelling that you mention here.
The tongue of the Mer de Glace has an increasing debris cover, which is until now restricted over the last kilometer downstream the Echelets gate. We assume this partially protected the front during the two last decades. While it is difficult to know how the debris cover will evolve above Echelets gate in the future we as conscious this could lead us to overestimated front retreat. The sensitivity experiment can this an insight into its influence.

Vincent Peyaud, on the behalf of the coauthors.

This future evolution can also be related to the glacier response time which has been a subject of interest of several studies. The response time of glaciers are affected by several predictors such as the glacier size, the glacier SMB and glacier slope which is the main driver. Recently, Zekollari et al. (2020) used large scale glaciers modelling to investigate the response time in the European Alps. For Mer de Glace the relation proposed by the authors gives a response time of 60 years, which is typical of the range of European Alps glaciers ($50 \pm 28$ years). In our sensitivity experiment the future simulation performed with values flux at Tacul and SMB stationary 
[revised manuscript text omitted]

---

## Author Response (AR1)

Dear Andreas Vieli, editor and dear reviewers,

Once again, thank you for your reviews and numerous comments. We regret that our answers were not clear enough. We clarify here our choices to include or not your suggestions in the attached manuscript.

Regarding your specific comments, please find below an answer to each of them:

2) some of the responses (in particular for referee 1) give rather lengthy explanations to the referees but they do not make clear how things have finally been addressed in this respect, I suspect nothing has been done to address it in the manuscript. So be clear in the response on what you have done or not done in the revisions (maybe not all planned revisions mentioned in the response have implemented in the track change document yet):
The revised document contains all our modifications.
These modifications are visible in the track change document we attached below. Minor changes are in blue; the removed text is kept as strikethrough text. New sections or paragraphs that were not implemented in the former track change document are visible in purple.
The pages and lines numbers refer to the track change document

2a) Explain briefly in the in an appropriate place in the methods part, why you used the flux gate method (lack of bed data, well constrains on flux etc….)
P. 5 L. 5 and P. 5 L. 13, We change the "boundary condition" section by introducing the gate boundary conditions of the model (at Tacul and Leschaux profile). Their parameterization is presented in the following. The next three sub-sections are dedicated to the other boundary conditions.

2c) Refer also in the main text (and maybe add some of the volume %/numbers results) to the additional new volume evolution figure in the supplements (R3-1) (see response to comment 2 of referee 3.
P. 11 L. 25, in the "Results" section, the area and volume evolutions are presented (with a reference to the Figures of the Supplementary Material) and underestimations are given.

2d) A clarification/adjustment in the manuscript may be needed on the point of the referee 1 (comment 6) of discrepancy between too thick and velocity too high. Currently it is not clear how this is addressed in the manuscript or just explained to the referee.
P. 13 L. 5, in the "discussion" part, we modify the text and include briefly our response to the referee #1 about the sentence "glacier is too thick and velocity too high".

2b) Mention briefly discuss your sensitivity analysis of the lapse-rate you present in the response to ref 1 and maybe add some figure detail to the supplement.
P. 14 L. 7, in the "discussion" part, we present the sensitivity experiments made on the vertical gradient in a new paragraph.

2e) …
Others clarifications have been included in the revised version:

P. 3 L. 4, we suggest to change the description of Mer de Glace in the "study site" section to clarify the distinction between the Mer de Glace glacier and the Géant glacier.

P. 9 L. 11, A sentence to explain the physical reason of the best fit of Tacul flux with the integrated surface mass balance upstream of the gate over the 11 previous years is presented.

P. 16 L. 28, We add a large paragraph in the discussion to present the specificity of our validation with the dataset and then compare our results with other studies.

P. 17 L. 7, We add a word on the sensitivity of our results to the value of the Glen law parameter that has not been explored.

We do not discuss more the influence of Leschaux gate as we explain (end of section 3.2.3) that its influence on Mer de Glace evolution is very limited.

3) the additional sensitivity analysis of the flux gate approach is now described at the end of the results (sect. 4.3) but the results are not really described/analysed in the results, there is just a reference to fig 8. And should the description of the experiments not rather be in the methods?
P. 17 L. 10, the earlier description of the sensitivity experiments was at the end of the results. A presentation of the main results has been added. Finally, we placed this paragraph, with the presentation of the main results, in the discussion "section" before we discuss their results.
We don't think this description belongs to "Methods section" that is long enough.

Readability of the figures.
The Figures are been improved, in particular we removed the parallel usage of green and red in Figs. 5 and 6. We homogenize the new choice of colors (fot the gates) in Figs. 1, 8 and S4.

English language.
We read with attention our manuscript, being vigilant to the English language.

Vincent Peyaud on the behalf of the coauthors

**Numerical modeling of the dynamics of Mer de Glace glacier, French Alps: comparison with past observations and forecasting of near future evolution.**

Vincent Peyaud[1], Coline Bouchayer[1,2], Olivier Gagliardini[1], Christian Vincent[1], Fabien Gillet-Chaulet[1], Delphine Six[1], and Olivier Laarman[1]

[1]Univ. Grenoble Alpes, CNRS, IRD, Grenoble INP, IGE, 38000 Grenoble, France
[2]Department of Geosciences, University of Oslo, 0316 Oslo, Norway

*Correspondence to:* Vincent Peyaud (vincent.peyaud@univ-grenoble-alpes.fr)

Small corrections in response to the reviews are in blue.
New section or paragraphs to take into account the suggestion of the reviews are in purple.

[revised manuscript text omitted]

---

## Author Response (AR2)

Dear Andreas Vieli,

We are glad our responses and the revised manuscript addressed satisfactorily the reviews and that the manuscript is acceptable for publication. We completed the requested modifications and address the corrected manuscript after a new careful proofread.

Below this letter, a version with the changes is attached. Most corrections do not request description but we have added few words for some points.

Something went wrong with the numbers/units of the mass balance gradients. I assume the should mostly be 0.009 /a (or 0.007 /a), but now it says 0.09 m/a /100m which would be 0.0009 /a. please check and correct throughout (p. 6 line 18+20, p. 13 and p. 15 (top).

For the unit of the vertical gradient we choose the notation $m\ a^{-1}\ (100\ m)^{-1}$ that is more intuitive than $a^{-1}$. Thank you for noticing the error in the values, we corrected it to *0.9 or 0.7 $m\ a^{-1}\ (100\ m)^{-1}$* at p. 6 line 20-23, p. 13 line 26 and p14. Line 3-5 of the marked-up manuscript version and in the supplementary material (Fig. S4 and its caption).

p.2, line 1: '…models…' (plural)

Done

p. 2 line 3: 'parameterizations' (plural. Same line, '…on a equilibrium ???

We changed the sentence and use the plural where needed:

« The first studies  of individual glaciers were restrained to flowline model**s** related to the local driving stress while studies on regional scale focused on empirical approach**es** in which ice dynamics is not explicitly taken into account and glacier evolution is based on parameterizations calibrated either on equilibrium-line altitude (ELA) model**s**, extrapolation**s** of observed geometry changes or volume and length–area scaling**s**. »

p. 10 line **19**: style '…that we slightly overestimated the thickness as well as the volume.'

Done

p. 10 line **20**: something wrong in this sentence and repeats 'underestimated' a bit too many times. Rephrase!

p. 11 line 2 to 5 of the marked-up version, this new paragraph was rewritten to improve its readability and remove repetitions:

« Figure 5 showed that hindcast thickness are slightly overestimated and this trend is also visible in the volume evolution. The volume decrease between 1979 and 2003 (14\% of the initial volume) is underestimated by 30\%. Extent reduction (10\% of the initial area) is also underestimated by 25\%. »

p. 18 line 19: not so clear what you refer to here with 'THIS future evolution…'. Is it 'This discrepancy in future evolution…'

p. 18 line 22: the sentence was not clear, it is modified to:

« The future evolution of the glacier can also be related to its response time which has been… »

Moreover, p. 9 line 15: we reversed references to Fig 5a and 5b content:

« centerline ice velocities (Fig. 5a) and observed surface elevation changes … (Fig.5b). »

And in Fig. 5, we added a) and b) in the subfigures and in the caption.

We sincerely acknowledge you and the three referees for your judicious comments and suggestions that improved the paper quality, a sentence was purposely added in the acknowledgement:

[revised manuscript text omitted]